# Noise-aware training of neuromorphic dynamic device networks

Luca Manneschi [1,5] ✉, Ian T. Vidamour [1,5] ✉, Kilian D. Stenning [2], Charles Swindells[1], Guru Venkat[1], David Griffin[3], Lai Gui[2], Daanish Sonawala[2], Denis Donskikh[2], Dana Hariga[1], Elisa Donati [4], Susan Stepney [3], Will R. Branford [2], Jack C. Gartside [2], Thomas J. Hayward [1], Matthew O. A. Ellis [1] & Eleni Vasilaki [1]

In materio computing offers the potential for widespread embodied intelligence by leveraging the intrinsic dynamics of complex systems for efficient sensing, processing, and interaction. While individual devices offer basic data processing capabilities, networks of interconnected devices can perform more complex and varied tasks. However, designing such networks for dynamic tasks is challenging in the absence of physical models and accurate characterization of device noise. We introduce the Noise-Aware Dynamic Optimization (NADO) framework for training networks of dynamical devices, using Neural Stochastic Differential Equations (Neural-SDEs) as differentiable digital twins to capture both the dynamics and stochasticity of devices with intrinsic memory. Our approach combines backpropagation through time with cascade learning, enabling effective exploitation of the temporal properties of physical devices. We validate this method on networks of spintronic devices across both temporal classification and regression tasks. By decoupling device model training from network connectivity optimization, our framework reduces data requirements and enables robust, gradient-based programming of dynamical devices without requiring analytical descriptions of their behaviour.

Before digital computation became widespread, analog dynamical systems were key in early computational platforms, with applications ranging from solving differential equations[1] to controlling early anti-aircraft guns[2]. These systems leveraged analogies between the inherent dynamic properties of analog components and their target applications to replicate behaviors with a high degree of control. They were particularly valuable for real-time testing beyond the capabilities of early digital computers, trading precision for speed[3]. However, as complementary metal-oxide semiconductor (CMOS) technology rapidly developed[4], digital platforms became increasingly fast and powerful. The greater accuracy and programmability of digital computers ultimately led to the replacement of analog systems by their digital counterparts.

More recently, the rapid expansion of machine learning has been propelled by the alignment between algorithms and hardware. Graphical[5] processing units (GPUs) and tensor[6] processing units (TPUs) have enabled the massive parallelization of matrix operations, leading to significant performance improvements by building large models out of relatively simple computational units. However, the increased reliance on large-scale models and extensive parallelization has also led to a worrying trend of rising energy costs[7].

*In-materio* computing, much like analog computing, harnesses the natural properties of materials to perform computations, providing an efficient alternative to conventional methods for data processing[8]. The principles of reservoir computing (RC)[9], which

[1]University of Sheffield, Sheffield, UK. [2]Blackett Laboratory, Imperial College London, London, UK. [3]University of York, Heslington, UK. [4]Institute of Neuroinformatics, University of Zurich and ETHZ, Zurich, Switzerland. [5]These authors contributed equally: Luca Manneschi, Ian T. Vidamour. ✉e-mail: l.manneschi@sheffield.ac.uk; i.vidamour@sheffield.ac.uk

originally involve fixed recurrent networks for computation, have been adapted to physical systems. In these systems, the inherent dynamics of the material serve as the computational resource[10–14]. In RC, only the output layer is trained, while the recurrent network–or, in the case of physical RC, the material–functions as a fixed temporal kernel, thus avoiding the complexities of optimizing dynamic processes. Reservoir computing has been explored in neuromorphic applications for temporal tasks like EMG classification in prosthetics[15,16], but its fixed dynamics limit adaptability and performance in complex tasks, highlighting the need for trainable systems that optimize parameters while accounting for noise and dynamics. Because the internal network structure itself is not trained, achieving the desired dynamic transformations often requires a high-dimensional network, as higher dimensions increase the likelihood of finding a suitable solution. Outputs from these systems can be obtained directly from the material itself[14], through multiplexing techniques[10], or by iteratively building networks by interconnecting multiple devices based on metric evaluations[17]. Despite these strategies, reservoir computing networks often face performance challenges when compared to networks where all parameters can be optimized using gradient-based methods. Fully optimisable networks typically perform better because they can adjust all their parameters to suit specific tasks[18].

To perform optimization on *in-materio* computers, general methodologies have been developed that train the interconnectivity of devices, leading to the concept of physical neural networks (PNNs)[19–21]. In PNNs, each node in a neural network corresponds to a physical device. Unlike neuromorphic computing platforms designed to closely emulate biological neural architectures or systems[22–26], PNN frameworks focus on optimizing the parameters that govern the interactions between devices. This approach allows for a flexible selection of material systems that offer a wide range of nonlinear responses, varying in complexity and functionality, akin to activation functions in artificial neural networks (ANNs).

Multiple approaches have emerged for optimizing PNNs. The Physics Aware Training (PAT) method[21] involves measuring device responses and estimating derivatives for backpropagation using a digital twin–a faithful model of the device. More recently, methods that avoid digital twins have been developed, using direct feedback alignment[19,27] or forward-forward algorithms[20,28] to optimize without gradient backpropagation. These methods approximate gradient descent with techniques directly applicable to the physical substrate, where devices provide simple transfer functions on current inputs. However, no existing approach can optimize PNNs in systems with dynamic behaviors and intrinsic memory–memory due to the inherent properties of the device materials–in general settings. Current methods assume devices are static and memoryless, and thus cannot optimize or leverage dynamic processes. As a result, they are unable to utilize functional memory sources, which are essential for temporally-driven tasks and *in-memory* computation. To fully harness material computational capabilities, a device-agnostic optimization method that accounts for dynamic processes is needed.

An important initial step in this direction was made with the proposal of using neural ordinary differential equations[29] (neural-ODEs) to model dynamic devices, with their feasibility demonstrated through simulations[30]. However, these models are not capable of capturing the noise in the system, which we hypothesize is essential for the robust transferability of parameters that control the interactions among devices from simulation to physical dynamical devices.

In this paper, we present the Noise-Aware Dynamic Optimization (NADO) framework, a universal framework for gradient-based optimization in deep networks of interacting dynamical systems. Our method does not require a mathematical description of the physical system, is entirely data-driven, and can be applied to any device that can be modeled as a differential equation, as long as sufficient sampling of input-output relationships is possible. To achieve this, we develop a generalized formulation of neural stochastic differential equations (Neural-SDEs)[31–33] capable of capturing colored noise, where different frequencies have varying power levels in the power spectral density, representing realistic noise characteristics observed in physical devices.

We apply our methodology to experimental spintronic devices previously used in neuromorphic computing applications[14,17,34–37]. This enabled performance in classification tasks beyond the capabilities of physical reservoir computing implementations using these systems, including in a gesture recognition task for generating motor commands for neuroprosthetic devices from real patients' surface electromyography signals. We demonstrate that noise modeling is crucial for transferring performance from simulations to networks of devices, allowing us to achieve high accuracy in regression tasks for the first time in fully optimized dynamic PNNs. Additionally, by employing cascade learning[38,39]–building the network layer by layer–we illustrate that, in principle, this methodology could be extended to arbitrarily deep networks, requiring only limited experimental data for each layer. This work marks a significant advancement in the application of complex material systems to PNNs, enabling gradient-descent-based, noise-aware optimization of the connectivity of arbitrary, mathematically-agnostic devices with intrinsic memory.

## Results

The NADO process for training networks of arbitrary dynamical devices involves three distinct phases, as illustrated in Fig. 1. First, differentiable digital twins–models that allow for the calculation of derivatives using standard tools–are trained to replicate the input-output responses of devices based on experimentally collected data (Fig. 1a). Next, these digital twins are used in network simulations of devices, where the interactions between the devices are optimized (Fig. 1b). Finally, the optimized parameters from the simulations are transferred directly to the physical network, where performance in benchmark tasks is assessed (Fig. 1c). An overview of these stages is provided below, with more detailed explanations available in the supplementary information.

### Neural-SDEs as differentiable digital twins

Previous work on fully-optimized PNNs has focused on devices without intrinsic memory. In contrast, networks of devices with intrinsic memory significantly increase training complexity because past inputs and states directly influence current behavior. This is analogous to a tennis player trying to hit a moving ball: any change in the position of the ball or the player affects all subsequent actions and movements (schematic in Fig. 2a). Similarly, in dynamical systems, adjustments made at any point in time can propagate through the network, affecting future states and complicating optimization. Algorithms that account for dynamic behaviors, such as backpropagation through time (BPTT)[40] and more recent methods[41,42], are crucial. These algorithms capture the influence of past changes on future states, enabling optimization over temporal sequences.

When computing gradients through numerical integration schemes–such as those used for training models based on differential equations–two methodological classes are typically considered: optimize-then-discretize (indirect) and discretize-then-optimize (direct). In direct methods, like backpropagation through time (BPTT) applied to the integration scheme, the continuous system is first discretized, and differentiation is performed on this explicit sequence of operations. This approach provides mathematical precision by ensuring that gradients are computed exactly for the numerical scheme being used, but requires storing all intermediate states, resulting in memory costs that scale as $\mathcal{O}(t)$ with the length of the input signal $t$. In contrast, indirect methods such as the adjoint sensitivity method[43] take the continuous-time gradients first, then discretize, which allows gradients to be computed with constant $\mathcal{O}(1)$ memory

a    Training digital twins

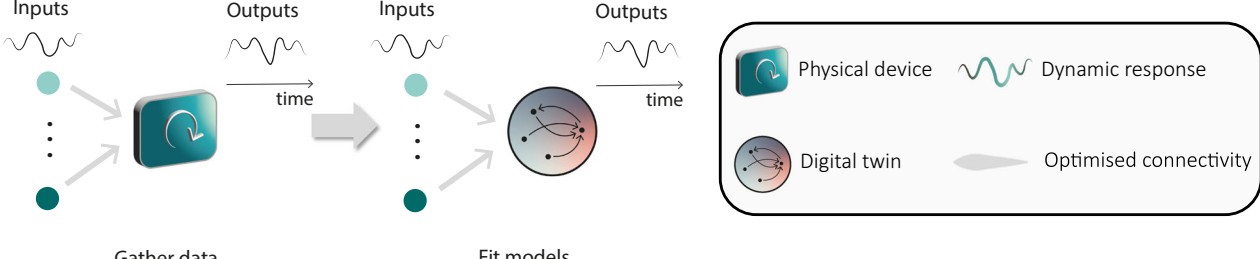

b    Simulated task optimisation                    c    Experimental task inference

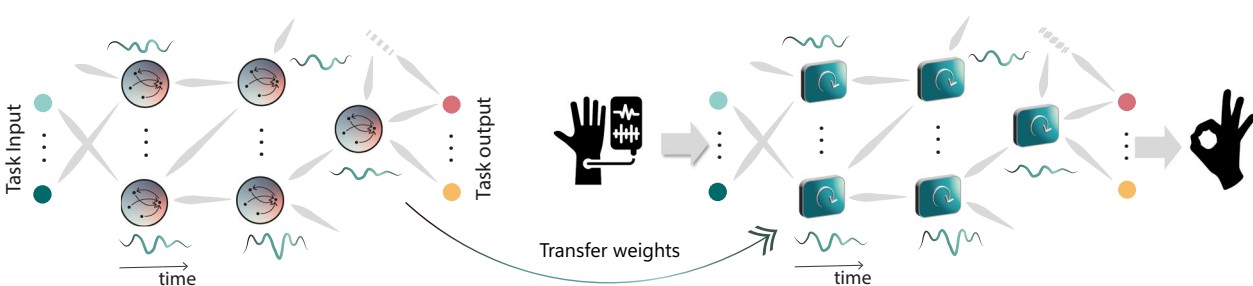

**Fig. 1 | Overview of the Noise-Aware Dynamic Optimization framework. a** Model Generation: Experimental devices (green squares) are driven under random inputs, their observable states are recorded, and these data are used to fit models of device dynamics (shaded circles). **b** Network Simulation: A neural network is constructed where each node replicates the dynamics of the original device, using the trained model. Parameters controlling device interactions (network weights) are optimized for a task via backpropagation through time (BPTT) or truncated-BPTT on the interacting digital twins. **c** Experimental Transfer: The parameters optimized in simulation are transferred like-for-like to experimental networks where each node is a real device, and task performance evaluated.

cost with respect to signal length. However, this can sometimes introduce additional numerical error or instability, depending on the integration method and system dynamics.

In this work, we employ direct methods to maximize accuracy, as they most faithfully represent the discretization performed, and it matches the optimization performed on the discriminator network in the GAN framework of the SDE (see Supplementary Fig. 2c). To manage the associated memory demands, we truncate gradients beyond the intrinsic memory length of the device, since contributions from longer histories are negligible. For devices with much longer intrinsic memory, the resource demands of direct methods may be impractical, making indirect approaches more attractive despite their potential trade-offs.

However, when applying these methods in practice, mismatches between the model and experimental data can accumulate over time, especially in deeper networks, where errors compound with each additional operation. Furthermore, physical systems rarely exhibit perfectly deterministic responses. Noise in experimental data cannot be captured by deterministic models alone, and optimization based solely on noise-free models often leads to sub-optimal solutions in real devices. By explicitly modeling noise in the simulation process, we close the simulation-reality gap and enable the discovery of network structures that are robust to noise, thereby improving the real-world performance of transferred networks.

In this work, we study two spintronic systems: nano-magnetic ring arrays (NRA)[35–37,44] and artificial spin vortex ice (ASVI)[14,17]. As illustrated in Fig. 2b for the NRA, device responses exhibit stochastic variation across repeated presentations of the same input sequence, with the response distribution shaped by both current and past inputs. This variability originates from intrinsic physical dynamics and experimental noise. To address the resulting simulation-reality gap, we extend the Neural stochastic differential equations (Neural SDE) framework to capture signal-dependent noise with complex autocorrelation. Compared to approaches such as long short-term memory

networks (LSTMs), neural differential equations provide several advantages: higher predictive accuracy, implicit access to partial derivatives via numerical integration, and natural integration of stochastic dynamics through SDEs. Supplementary Fig. 16 compares the prediction accuracy and number of trainable parameters for Neural ODE and LSTM models, and information on dataset construction and hyperparameter selection are tabulated in Supplementary Fig. 17.

Figure 2c shows the architecture of the proposed Neural-SDE model, comprising two neural networks: one for deterministic dynamics and another for stochastic dynamics. These networks are coupled via the numerical integration method, enabling the model to represent explicitly how system output depends on device state and external inputs, incorporating both deterministic and stochastic components. This structure supports noise-aware gradient computation for backpropagation through time (BPTT). The Neural-SDE architecture thus parameterizes the stochastic differential equations that define how device output evolves as a function of the current state and external input.

The deterministic network (upper) and the stochastic network (lower) each receive external input signals and a sequence of past device states. The history length must be sufficient to approximate the system as Markovian, ensuring that future states can be predicted from current inputs and recent device states. When this criterion is met, the method is applicable to any dynamical system. The stochastic network also receives auxiliary variables to support noise modeling. Outputs from both networks are integrated using a stochastic numerical scheme to produce the device's activity (readout) at the next timestep. This value is recursively fed back as the most recent device state. Orange arrows in Fig. 2c indicate error gradients with respect to device activity and external input, as computed during BPTT.

Figure 2d compares the outputs of the two models. The neural-ODE model provides deterministic predictions that capture the main trend for given inputs. In contrast, the neural-SDE model generates a

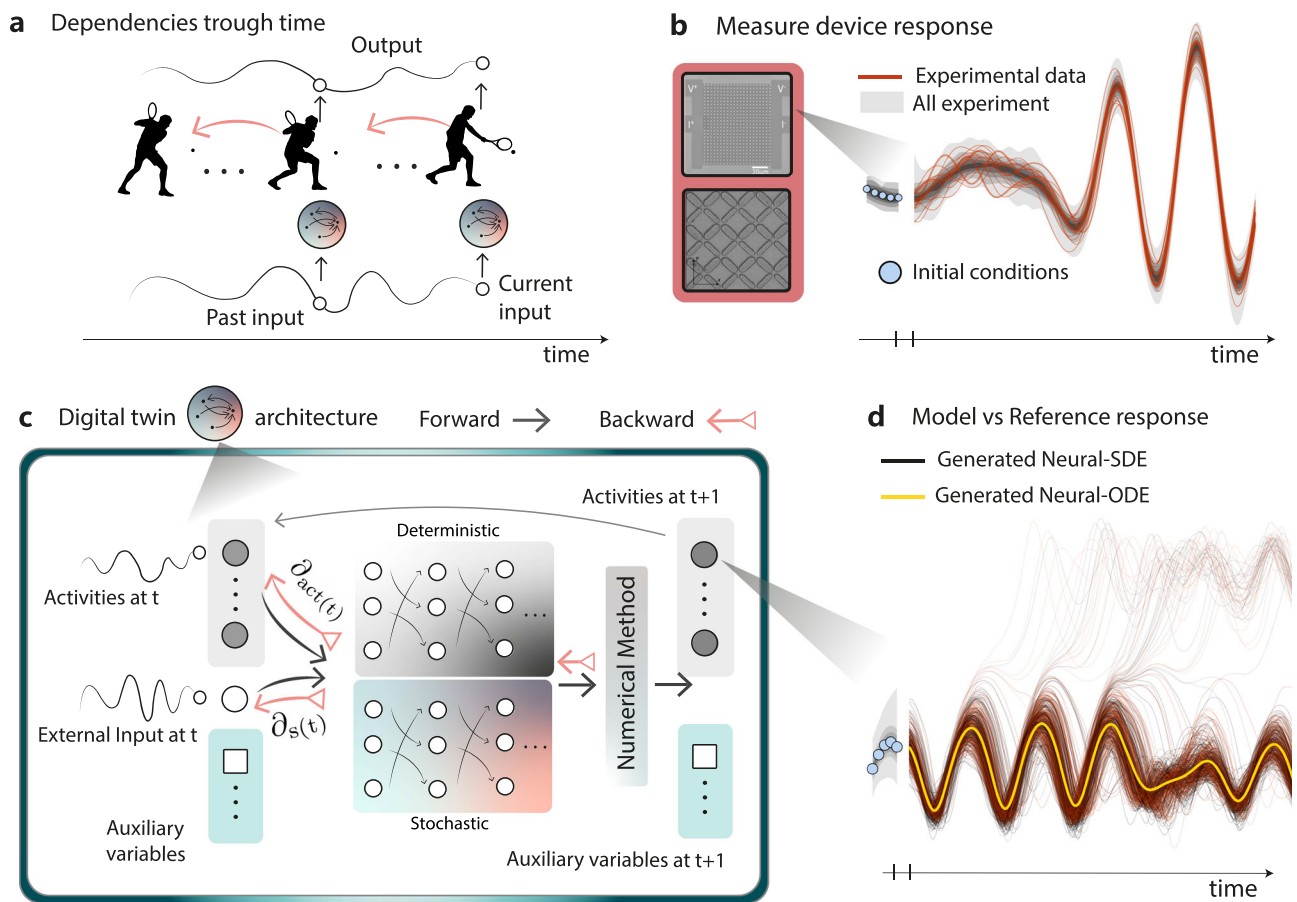

**Fig. 2 | Modeling and optimizing dynamic behaviors. a** Schematic analogy of temporal dependencies. Altering an action in the past has consequences for all future actions. Similarly, for backpropagation through time (visualized by orange arrows), changes to the final output caused by all past inputs and states must be taken into consideration. In the framework, digital twins (shaded circles) are used to estimate dependencies between output and input. **b** Schematic showing samples from distributions of initial conditions, which subsequently affect the predicted trajectory. Gray clouds show the distribution of all gathered data for a given random input sequence, while red lines highlight specific trajectories. **c** Schematic diagram of the Neural-SDE architecture. Inputs of device states (activities), external driving stimuli, and auxiliary variables feed into a pair of distinct neural networks that handle the deterministic (upper network) and stochastic (lower network) behaviors. The output of these networks feeds into a numerical ODE solver, generating predictions of both activities and auxiliary variables for the next timestep.

The results are recursively fed back as inputs to the next timestep prediction, generating predicted trajectories from initial conditions and external driving signals. Black arrows show forward propagation of activities; orange arrows show backward propagation of gradients. **d** Comparison between predictions generated via neural-ODE and neural-SDE models. The neural-ODE produces a single deterministic outcome for a given set of initial conditions and input stimuli, shown by the yellow line. The neural-SDE instead generates sampled trajectories from a distribution based on the learned noise characteristics. The black lines show 100 generations of a signal via the neural-SDE, while red lines show real experimental data from repeated identical input sequences. As in (b), blue circles represent selected initial conditions and the gray clouds represent the distributions observed across all experiments. Further comparisons of experimental versus simulated trajectories of both experimental devices can be found in ?

range of trajectories that approximate the observed distributions for the same inputs. The reference dynamics (red lines) show noise-induced bifurcations, which are not captured by the neural-ODE model but are effectively modeled by the neural-SDE model.

## Temporal classification benchmarks

Here, we use the NADO framework to optimize the connectivity in networks of interacting devices. The physical devices represent each hidden node in the network and are treated as fixed nonlinear temporal kernels, with only the weights of the network optimized. First, we perform a classification task on a modified version of the MNIST dataset. To introduce a memory component, each MNIST digit image was split into $n$ separate images containing a random subset of the original pixels. When combined, these images reconstruct the full digit. The partial images were presented to the physical system sequentially, requiring it to use memory to classify the original digit from the sequence. Final predictions were based on the network's response at the end of the sequence.

To extend our study to a more challenging task, the NRA networks were trained to recognize hand gestures for controlling a neuroprosthetic device[45]. This task used real-world electromyography (EMG) data collected from the forearms of patients performing seventeen different hand and wrist movements. Predictions were based on the class with the highest output within a window corresponding to data acquired between 120-180 ms after the gesture onset (see Supplementary Fig. 13). The integration of neuromorphic systems with EMG data presents a promising avenue for addressing the challenges of real-time temporal classification[46,47], with the potential to leverage low-energy computation to improve the efficiency of gesture recognition for neuroprosthetic applications.

Figure 3a provides an illustration of the tasks. Figure 3b compares the responses of the physical network of magnetic nanorings to their simulated counterparts. The dynamics of four nodes from two different layers are shown in red. Simulated activities are in black, with the digital twins' response distribution in gray. The horizontal bars represent output activations of different physical devices for a given

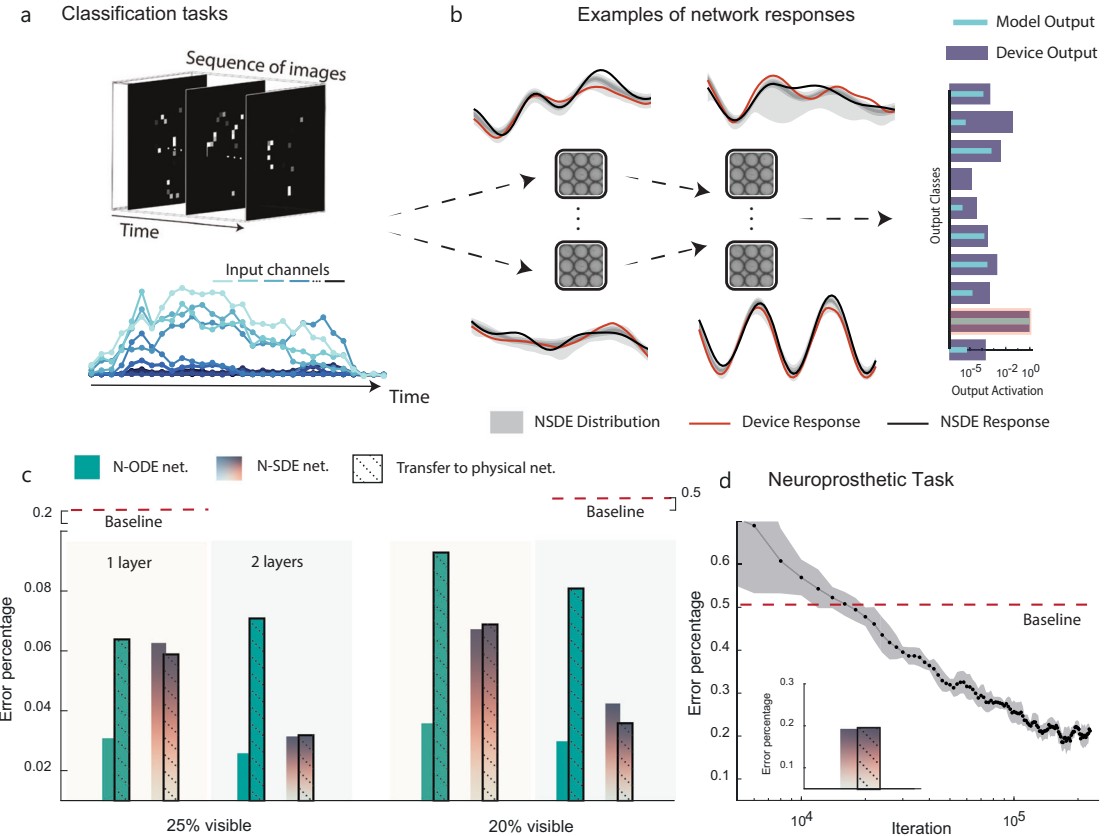

**Fig. 3 | Partially-observable MNIST and Neuroprosthetic movement classification tasks. a** The MNIST data, presented as sequences of images, have been adapted into a temporal problem by partially obscuring the images at each time step, requiring the system to integrate information over time for accurate classification. The neuroprosthetic gesture recognition task is characterized by input channels that vary over time. **b** Example responses from the network's physical nodes, showing experimentally measured responses (red) and digital twins' responses (black) for different nodes across two layers. The gray areas represent the distribution of responses from the digital twins, while the dashed arrows illustrate the flow of information from the input through the layers to the output. The horizontal bars indicate the output activations of experimental networks (dark blue) compared to the model output (light blue), with the correct class highlighted in red. **c** Transferred performance of nanoring array networks using Neural-ODEs (green bars) and Neural-SDEs (orange bars) as digital twins in the MNIST benchmark. The deterministic Neural-ODE models exhibit unrealistically high performance in simulation (no border), which significantly deteriorates in experiments (black border). In contrast, the noise-aware training provided by Neural-SDEs maintains high performance on physical devices, demonstrating effective exploitation of node dynamics and robustness during device transfer. **d** Performance of the Neural-SDE models on neuroprosthetic gesture recognition, demonstrating the framework's potential in addressing real-world tasks. The black line represents the error as a percentage across iterations. The inset shows the final performance, comparing simulation results with those after transfer to the physical device. Details on hyperparameter selection can be found in 17.

input frame. Despite some mismatches, the simulated network's activities correlate well with the experiment, leading to the same classification outcome. Figure 3c displays the predicted and experimentally achieved accuracies for different task difficulties, measured as the percentage of visible pixels, in networks with single and two hidden layers. The task proved challenging for networks with a single hidden layer due to the well-established non-linearity/memory trade-off[48], where a single hidden layer is tasked with both remembering past inputs and non-linearly combining the information simultaneously. This challenge is also observed in analytical systems, where performance significantly improves when the network architecture includes more than one layer (see Supplementary Information).

To demonstrate the importance of noise awareness in network optimization, we compare Neural-ODEs and Neural-SDEs as digital twins (Fig. 3c). Neural-ODEs, which do not incorporate noise, fail to provide information about noisy response regions that should be avoided during optimization. This limitation results in inaccurate predictions of the physical neural network's performance, particularly when additional hidden layers are added. In contrast, Neural-SDEs account for noise, enabling the optimization process to identify parameter values that remain robust under stochastic variations and

experimental conditions. These findings underscore the necessity of incorporating noise into digital twins to achieve reliable network optimization.

The baseline performance, shown as red dashed lines, corresponds to physical neural networks with the same architecture but randomized hidden-layer connectivity, following the reservoir computing paradigm for both one-layer (left, orange) and two-layer (right, green) networks. The hidden weights are randomly drawn from distributions matched to those of the Neural-SDE-optimized network (see Supplementary Fig. 11b), while only the output weights are optimized. This demonstrates that achieving high performance requires optimization of the entire network connectivity, not just the output layer.

Two-layer networks of nanorings optimized using interacting neural-SDEs demonstrate accurate transfer to physical devices and achieve performance exceeding that of static software-based MLPs with identical architectures (two hidden layers of 200 nodes). These networks also match the performance of dynamic MLPs incorporating leaky integrators (transferred accuracies for NRAs: 96.8% at 25% visibility and 96.0% at 20% visibility, versus 90.0% and 88.0% for static MLPs, and 96.9% and 96.7% for leaky-integrator MLPs; see Supplementary Fig. 10). These results show that the magnetic nanoring PNN

effectively exploits the intrinsic device dynamics as a memory resource, underscoring the capability of the proposed framework to leverage physical dynamics for in-memory computing. However, the limited memory of NRAs results in a relative decline in performance at lower visibility compared to software leaky integrator networks.

To demonstrate applicability to real-world problems, we applied the NADO approach to a neuroprosthetics task[45], where the goal was to classify human gestures using surface electromyography data from ten forearm electrodes. This task is substantially more challenging than the partially observable MNIST benchmark due to longer input sequences accumulating more experimental noise, and the need for generalization across both gestures and different subjects.

Figure 3d shows the measured performance of a network whose connectivity was optimized in simulation using digital twins and then transferred to the physical system. As a baseline, we used a randomly connected network with a single hidden layer and trained only the output weights, following the standard reservoir computing (RC) approach, with the number of nodes matched to the PNN. This adjustment from previous baselines avoids the performance degradation seen with random connections in deeper networks while maintaining the same node count. The optimized network achieves an error rate approximately 30% lower than the baseline, demonstrating the advantages of optimized connectivity over conventional *in-materio* RC and highlighting the improved capability made possible by connectivity optimization.

### Regression benchmark

Classification problems are generally more forgiving when transferring parameters, as their winner-takes-all algorithm only requires the highest activation in the correct class for accurate prediction. In contrast, regression problems demand specific, continuous output values from the network, presenting a more challenging task when transferred to physical networks. To test the limits of our methodology, we applied the network to predict the Mackey-Glass system operating in a regime characterized by quasi-periodic, chaotic behaviors. This network comprised a mixture of NRA and ASVI nodes. The network structure was designed to leverage the distinct characteristics of each physical system: the first layer featured ASVIs, which exploited their high output dimensionality to project the low-dimensional Mackey-Glass signal into a higher-dimensional space, while the subsequent layers consisted of NRAs to provide nonlinearity and memory for learning the underlying dynamics.

To mitigate error accumulation with an increasing number of layers, which is more pronounced in regression problems, we employed the cascade-correlation algorithm[38], adapted for experimental settings. A schematic of this process is shown in Fig. 4a. In this approach, hidden layers were trained sequentially, with previously trained parameters remaining fixed. As a result, the learning process treats the responses of earlier layers as fixed inputs for the layer currently undergoing optimization. Once a layer was optimized, experimental node activities were gathered using the learned parameters, generating 'ground-truth' data with zero mismatch in the forward pass for the subsequent layer to be trained. This strategy helped to correct the digital twin's simulation-reality gap, which might otherwise be amplified throughout the network depth. Cascade-correlation, therefore, limits the propagation of errors to a single layer, facilitating better transfer. In this respect, the methodology is similar to Physics-Aware Training[21], but requires only one epoch of data per device in the physical neural network structure, rather than continuous sampling during each iteration (For further proofs on error bounding and potential limitations of the approach, see Supplementary Information 'Analysis of Cascade Learning Approach').

Figure 4b shows the mean squared errors (MSE) between the ground truth dynamical equations and the predictions as a function of prediction steps into the future, for transferred networks with two

(red) and three (blue) hidden layers. Cascade learning achieves the lowest MSE, indicating excellent alignment between target and prediction, as illustrated in Fig. 4c. For reference, previous implementations of these experimental systems on this task reported a peak MSE of $3.86 \times 10^{-2}$ at t+5 using the reservoir computing paradigm[14], and approximately $1 \times 10^{-2}$ with multilayer PNNs trained without gradient-based optimization[17].

Without the corrective dataset, performance deteriorates as the network depth increases. However, retraining with the corrective dataset between layers reduces overall error, and adding more layers improves performance. This outcome highlights the scalability of the methodology, enabling the construction of deeper networks while minimizing additional data collection. By confining mismatch error to a single layer, this approach can be extended to create arbitrarily deep dynamic PPNs. However, as in any machine learning network, an improvement in performance is not guaranteed by adding additional hidden layers, and the configurations learned via cascade learning may be sub-optimal when compared to full optimization in the absence of simulation-reality mismatch, with techniques such as PAT[21] serving as useful methodologies for minimizing this gap where experimental throughput allows reasonable training times (see Supplementary Figs. 8 and 9).

When compared to digitally implemented, noiseless dynamical neural networks (see Supplementary Fig. 8), the hybrid physical neural networks incur additional error. When predicting 5 steps ahead for the Mackey-Glass future prediction tasks, three hidden-layer networks of simulated leaky integrators achieve mean MSEs $1 \times 10^{-3}$ compared to $2.2 \times 10^{-3}$ for physical networks trained via the same cascade-learning approach. However, this is likely due to the effect of experimental noise impacting prediction of noiseless, mathematically-defined target signals. In spite of this, the significant improvement compared to previous implementation of this task using the same devices as highlighted earlier highlights the promise of the NADO approach.

## Discussion

Devices with complex dynamical responses are powerful substrates for the physical implementation of neural networks designed for temporal processing. While individual devices may possess limited computational capacity, learned connectivity within device networks brings them closer to the performance of deep artificial neural networks. This work highlights the critical role of the Noise-Aware Dynamic Optimization (NADO) framework in optimizing connectivity within dynamical physical neural networks. Central to this framework is the development of stochastic digital twins based on the neural SDE approach.

These models are differentiable and provide surrogate gradients for task-specific, gradient-based network optimization. Notably, the NADO framework requires no prior knowledge of the underlying system and minimizes physical device usage during training–accelerating the optimization process in cases where data acquisition is slow (see Supplementary Fig. 9). We demonstrate the effectiveness of this framework by successfully training networks of complex physical neurons to solve a range of temporal tasks: partially observable MNIST classification, forward prediction of the Mackey-Glass sequence, and gesture recognition for a neuroprosthetic device.

Previous methods for training physical networks, such as Physics-Aware Training (PAT)[21] and Physical Local Learning (PhyLL)[20], have been limited to static devices. In contrast, our framework embraces the dynamical nature of physical systems, treating this complexity not as a hindrance but as a computational asset. To our knowledge, this is the first demonstration that interconnected physical devices can be optimized using backpropagation through time (BPTT), the foundational learning algorithm for recurrent neural networks and dynamical systems.

Our use of digital twins shares conceptual parallels with PAT but extends to dynamically driven, noise-aware models. Whereas PAT relies on experimental measurements to correct a model's internal

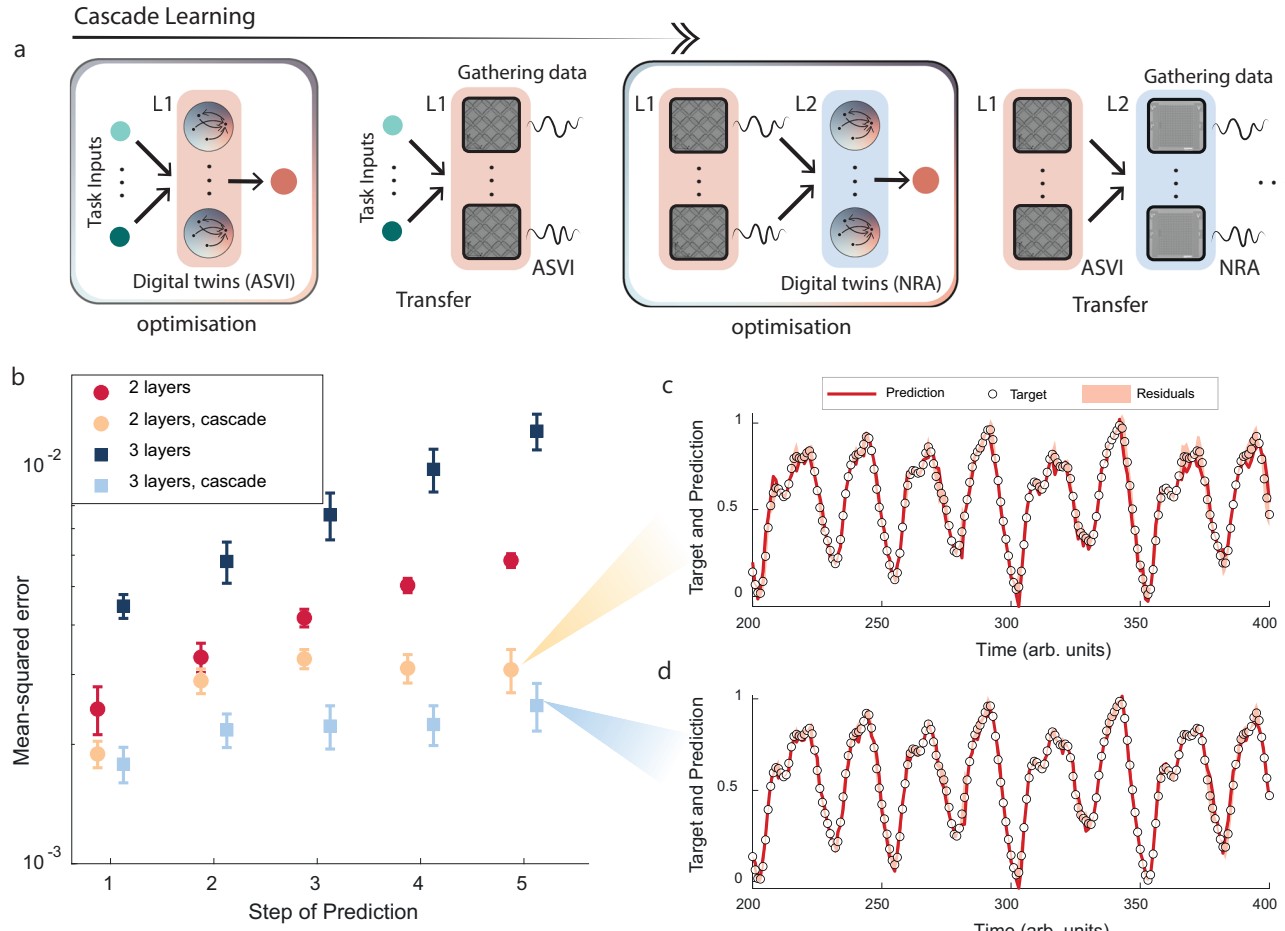

**Fig. 4 | Cascade learning and Mackey–Glass future prediction task. a** Schematic overview of the methodology employed for sequentially training network layers with intermediate data gathering. The boxes represent steps performed in simulations, with red shading indicating ASVI twins/experiments in the first layer (L1) and blue shading representing NRAs in the second layer (L2). Initially, a single ASVI layer is connected to a simulated output neuron and trained for the regression task. Once trained, the connectivity from the input to the ASVI layer is transferred to the physical device. Experimental data is then collected to serve as input for training the connectivity to the subsequent layer, consisting of NRAs. This process can, in principle, be extended to accommodate any number of layers. Retraining the digital twin is not required; intermediate data are used solely to adjust the connectivity between the new and the previous layer. **b** Mean-squared error between

ground truth and experimental network predictions for the Mackey-Glass future prediction task as the number of future steps increases. red circles/ blue squares represent networks with two/three hidden layers, while dark/light colors compare direct training of the entire network to networks trained using cascade learning, as presented in (**a**). Error bars show standard deviations of error over 10 different subsections of 4000 test samples. Comparison between model prediction and ground-truth data for the five-timestep future prediction of the Mackey-Glass equation in (**c**) two-layer and (**d**) three-layer networks. White circles represent the ground truth data, red lines show the transferred PNN prediction, and pink shading indicates the difference between the ground truth and the network prediction. Details on hyperparameter selection can be found in 17.

activity and reduce the simulation-reality gap during training, our neural SDE framework enables effective optimization without requiring real-world data for optimization in classification tasks.

Extending PAT to the dynamical setting involves correcting the system's computational graph through time (see Supplementary Information - Generalizing PAT to Dynamical Settings). This requires sampling device states across temporal trajectories and adjusting estimates of both system state evolution and input-output dependencies. In the Supplementary Information, we detail this generalization and evaluate performance as a function of the adopted sampling strategy. Notably, sampling every device at every time step for each input signal during training is experimentally demanding, and practically infeasible with the hardware considered here, due to relatively slow experimental throughput. This limitation is compounded by the inability to parallelize across batch sizes in hardware.

Nevertheless, PAT-inspired sampling strategies remain valuable for refining model behavior. Our neural SDE framework should not be seen as a replacement, but as a complementary alternative. There is no

intrinsic barrier to applying experimental corrections to neural SDE models to further reduce this gap. For the regression task investigated, we employed a cascade learning approach–interpretable as a sparse variant of PAT–to incrementally correct neural SDE activity layer by layer. This enabled us to balance theoretical performance with experimental feasibility. As observed in regression tasks, some degree of sampling was necessary; however, satisfactory performance was achieved without continuous correction through time or at every parameter update.

Adopting neural SDEs as the underlying physical model provides a robust mechanism for generating noisy samples during digital training and yields a differentiable representation of device stochasticity. This enables gradients to be backpropagated through the stochastic component itself–a critical feature for systems whose responses depend non-trivially on specific noise realizations, which PAT alone cannot account for (see Supplementary Fig. 10).

While the dynamical systems studied here are promising candidates for neuromorphic computing, current approaches to signal

input and state readout present practical challenges for large-scale implementation. For example, artificial spin-vortex ice requires high-precision, low-throughput measurement equipment, and applying magnetic fields is both slow and energy-intensive. Similarly, nanoring devices rely on electromagnets that consume much more energy than the underlying physical computation, and bridging electrical and magnetic domains adds further complexity. Nonetheless, the methodology described here is broadly applicable to any dynamical system modeled by differential equations. This flexibility enables the optimization of networks based on alternative device platforms, where integration with existing CMOS technology may be more straightforward.

Additionally, the physical neural networks used here are constructed by serially sampling the same device of each class. Although the ensemble nature of the magnetic devices here leads to limited device-to-device variability when manufactured with nominally identical design processes, the additional complexity of device-to-device variability is not directly considered here. For limited device variablity, the inherent noise-awareness of the optimization approach will mitigate noise from device variability, provided that a representative number of devices are used to train aggregate neural-SDE models that reproduce both the mean behavior and predicted variance across devices. However, the increased variance in behavior may result in lower peak performances.

For the first time, we demonstrate the effectiveness of neural SDEs with an extension to include colored noise on experimental data from neuromorphic systems. Our framework generalizes across device types, requiring only that both deterministic and stochastic dynamics can be sampled. While prior work has explored neural ODEs in spintronic simulations[30], we apply this approach to two distinct experimental platforms: nanomagnetic ring arrays and artificial spin vortex ice, with particular emphasis on capturing their intrinsic stochasticity.

This work introduces a unified optimization framework that enables joint training over input signals and device parameters, allowing precise control over the operational regime of physical neural networks. Although developed in a neuromorphic context, the method is broadly applicable to systems where learning directly from real-world dynamical processes is critical.

## Methods

### Sampling device behaviors
For each of the device classes, a single device was repeatedly sampled. First, the range of inputs at which the devices are dynamically active was established by sweeping input stimuli and observing changes in measured output. Data used for training the models of dynamic behaviors were sampled randomly from the determined input range. Different datasets were constructed for training the deterministic model and stochastic model behaviors. In both cases, the systems are initialized by a strong pulse of magnetic field, saturating the devices. A single input corresponding to the maximum allowed input value is then applied, generating a trajectory to be used for initial conditions of the model. For the Neural-ODE, devices were then driven by many uncorrelated, randomly generated input sequences sampled from a uniform distribution spanning the range of activity, with the measured state of the devices recorded alongside the external input at each time. To gather a validation set for the optimization of the neural-SDE, the devices were driven by 100 repetitions of each sequence from a smaller set of randomly sampled sequences, with similar recordings of input and measured state.

### Neural-ODE modeling
The Neural-ODE models used here emulate the observable state of a dynamical system $\mathbf{x}(t)$, which is an $N_x$-dimensional vector gathered experimentally. This is done by parameterizing the instantaneous gradient of the dynamical systems with respect to its current hidden state $\mathbf{y}(t)$ and external input $\mathbf{s}(t)$ via a neural network $\mathbf{f}$, before integrating to

find the next state. This process is described in further detail below, with a didactic tutorial provided in the Supplementary Information.

As in ref. 30, the unknown internal state of the system considered is embedded by concatenating a set of delayed observables $\mathbf{x}(t)$ to $\mathbf{x}(t - N_{delay}\delta t)$, where $N_{delay}$ is the number of delays adopted. We define this augmented $N_x(N_{delay}+1)$-dimensional state as $\mathbf{y}(t) = \left(\mathbf{x}(t), \ldots, \mathbf{x}(t - N_{delay}\delta t)\right)$. Assuming this representation renders the system Markovian for a given $\mathbf{s}(t)$ and $\mathbf{y}(t)$ so that a dataset device dynamics, $\mathcal{D} = \{(\mathbf{s}(0), \mathbf{y}(0), \mathbf{y}(1)), \ldots, (\mathbf{s}(t), \mathbf{y}(t), \mathbf{y}(t+\delta t)), \ldots\}$ can be sampled for training the model to predict the next step.

To predict trajectories, the Neural-ODE is provided with initial conditions $\mathbf{y}(t_0)$ sampled from a random starting time $t_0$. Then, driven by external signals $(\mathbf{s}(t_0), , \ldots, \mathbf{s}(t_0 + T\delta t))$, the model is asked to predict the evolution $(\mathbf{y}(t_0 + \delta t), \ldots, \mathbf{y}(t_0 + T\delta t))$ of the system for T steps. We will denote the predicted activities generated by the model as $\tilde{\mathbf{y}}(t)$ to differentiate them from the target $\mathbf{y}(t)$. The neural-ODE activities are defined through the integration of

$$d\tilde{\mathbf{y}}(t) = \mathbf{f}\left(\tilde{\mathbf{y}}(t), \mathbf{s}(t), t | \phi^f\right) dt \qquad (1)$$

with $\tilde{\mathbf{y}}(t_0) = \mathbf{y}(t_0)$, and the neural network $\mathbf{f}(\cdot | \phi^f) : \mathcal{R}^{N_x(N_{delay}+1)+N_s} \rightarrow \mathcal{R}^{N_x(N_{delay}+1)}$, parameterized by weights $\phi^f$, estimates the instantaneous gradient of the system from $\tilde{\mathbf{y}}(t)$ and $\mathbf{s}(t)$. Integration via numerical methods leads to the prediction of $\mathbf{x}(t + \delta t)$ as the most recent state in $\tilde{\mathbf{y}}(t + \delta t)$. This iterative, recursive process continues for all steps considered $t \in [t_0, t_0 + T\delta t]$. The network is trained by minimizing the mean squared error between model generated states $\hat{\mathbf{y}}$ and experimentally gathered states $\mathbf{y}$. We refer to the Supplementary Information for further details and training hyper-parameters.

### Neural-SDE modeling
The Neural-SDE model proposed captures both the stochastic[49] and deterministic behaviors of dynamical systems, as well as noise in experimental measurements. The model features an additional network $\mathbf{g}$ to the Neural-ODE model introduced in the previous section. To account for various experimental settings, the neural-SDE has been designed to accommodate external signals, delayed observations of systems activities, and the presence of colored noise. The incorporation of colored noise involves the introduction of $N_a$ auxiliary variables $\mathbf{a}(t)$ operating over multiple timescales. The proposed neural-SDE model is defined as

$$\begin{pmatrix} d\tilde{\mathbf{y}}(t) \\ d\mathbf{a}(t) \end{pmatrix} = \begin{pmatrix} \mathbf{f}(\tilde{\mathbf{y}}(t), \mathbf{a}(t), \mathbf{s}(t), t | \phi^f) \\ -\boldsymbol{\tau}^{-1}\mathbf{a}(t) \end{pmatrix} dt + \mathbf{g}(\tilde{\mathbf{y}}(t), \mathbf{a}(t), \mathbf{s}(t), t | \phi^g) d\mathbf{W} \qquad (2)$$

with $\tilde{\mathbf{y}}(t_0) = \mathbf{y}(t_0)$ and $\mathbf{a}(t_0) = \mathbf{0}$ as initial conditionas and $d\mathbf{W}$ is a $N_W$-dimensional derivative of the Wiener process[50] that during simulation is given by Gaussian random numbers. The functions $\mathbf{f}(\cdot | \phi^f)$ and $\mathbf{g}(\cdot | \phi^g)$ are neural networks defined by the trainable weights $\phi^f$ and $\phi^g$ respectively which are learned during training. $\boldsymbol{\tau}$ is a diagonal matrix defining the timescales of the auxiliary variables are also learned during training.

We generally found that it is important to restrict the function $\mathbf{g}$ to generate stochasticity only on the most recent prediction of $\tilde{\mathbf{x}}$. By expanding out the augmented $\mathbf{y}(t)$ vector and introducing the restriction on the $\mathbf{g}$ function leads to redefinition of Eq. (2) as

$$\begin{pmatrix} d\tilde{\mathbf{x}}(t) \\ d\tilde{\mathbf{x}}(t - \delta t) \\ \vdots \\ d\tilde{\mathbf{x}}(t - N_{delay}\delta t) \\ d\mathbf{a}(t) \end{pmatrix} = \begin{pmatrix} \mathbf{f}_1(\tilde{\mathbf{y}}(t), \mathbf{s}(t), t | \phi^f) + \mathbf{A}\mathbf{a}(t) \\ \mathbf{f}_2(\tilde{\mathbf{y}}(t), \mathbf{s}(t), t | \phi^f) \\ \vdots \\ \mathbf{f}_{N_{delay}+1}(\tilde{\mathbf{y}}(t), \mathbf{s}(t), t | \phi^f) \\ -\boldsymbol{\tau}^{-1}\mathbf{a}(t) \end{pmatrix} dt + \begin{pmatrix} \mathbf{g}_1(\tilde{\mathbf{y}}(t), \mathbf{s}(t), t | \phi^g) \\ \mathbf{0} \\ \vdots \\ \mathbf{0} \\ \mathbf{g}_a(\tilde{\mathbf{y}}(t), \mathbf{s}(t), t | \phi^g) \end{pmatrix} d\mathbf{W}$$

$$(3)$$

where the dimensionality of the function $\mathbf{f}(\cdot|\boldsymbol{\phi}^f) : \mathcal{R}^{N_x(N_{delay}+1)+N_s} \rightarrow \mathcal{R}^{N_x(N_{delay}+1)}$ is analogous to the neural-ODE settings, while the non-zero elements of the function $\mathbf{g}(\cdot|\boldsymbol{\phi}^g)$ constitute a map $\mathcal{R}^{N_x(N_{delay}+1)+N_s} \rightarrow \mathcal{R}^{(N_x+N_a)\times N_W}$. $\mathbf{A}$ is introduced as a $N_x \times N_W$ dimensional matrix linking the auxiliary variables to the last state of the system. We highlight that a single neural network mapping $\mathcal{R}^{N_x(N_{delay}+1)+N_s} \rightarrow \mathcal{R}^{(N_x+N_a)\times N_W}$ was used to jointly parametrize the two terms $\mathbf{g}_1$ and $\mathbf{g}_a$ as depicted in the schemes of the neural-SDE architecture reported in Fig. 2. The training of the neural-SDE follows the generative adversarial network paradigm introduced in previous works[49] and further details are provided in the Supplementary Information.

## Noise-aware dynamic optimization of physical neural networks

We now consider of a specific network architecture, formed of physical nodes modeled as Neural-SDEs. Here, we assume a multilayer perceptron (MLP) structure composed of hidden layers indexed by $l = 1, ..., L$ with the $i-th$ node in each layer having an augmented state $\mathbf{y}_{l,i}(t)$. This augmented state was defined to describe the device as a Markovian system and consequently to capture the system behavior in a neural-ODE/SDE framework. As a consequence, it is a modeling abstraction and does not correspond to the information exchanged between devices when embedded in a physical neural network. Therefore, we assume device-to-device interactions to occur via the variable $\mathbf{y}_i^\pi(t) = \boldsymbol{\pi}(\mathbf{y}_i(t))$ where $\boldsymbol{\pi}$ is a post-processing function providing a mapping of the device dynamics to the quantities that dictate the exchange of information. Importantly, this function needs to be differentiable to permit backpropagation through the network dynamics.

Within our proposed NADO framework, the key goal is to optimize the weights connecting each of these physical nodes and we fix the parameters governing the neural-ODE/SDE mdoels. To simplify the notation we define $\mathbf{z}_l^\pi(t)$ as the concatenated activities of all nodes within layer $l$, such that $\mathbf{z}_l^\pi(t) = \left(\mathbf{y}_{l,1}^\pi(t), \mathbf{y}_{l,2}^\pi(t), \dots\right)$ and $\mathbf{S}_l(t) = (\mathbf{s}_{l,1}(t), \mathbf{s}_{l,2}(t), \dots)$ is the concatenated input to layer $l$. As before, a tilde $(\tilde{\cdot})$ is adopted to distinguish the experimental from the simulated quantities and so we define $\tilde{\mathbf{z}}_l^\pi(t)$ as the simulated analog of concatenated $\tilde{\mathbf{y}}_{l,i}^\pi(t)$ digital twin dynamics. During a forward pass of the simulated network, we initialize each digital twin's internal state $\tilde{\mathbf{y}}(t_0)$ using a distribution over initial conditions, which captures the inherent stochasticity of the inital device states $p(\mathbf{y}(t_0))$. As described in the Supplementary Information, this distribution is empirically measured rather than learned, in order to reduce discrepancies between simulation and real-world behavior. Under this setup the input to layer $l + 1$ can be written as

$$\mathbf{S}_{l+1}(t) = \mathbf{h}\left(\boldsymbol{\theta}_{l+1}\mathbf{z}_l^\pi(t)\right) \tag{4}$$

where $\boldsymbol{\theta}_{l+1}$ represents the weight matrix connecting layer $l$ to $l+1$ and $\mathbf{h}$ is a device pre-processing function that maps the raw inputs into a physical input for each device in the layer, for example converting into an appropriate magnetic field value. Further details and extension are provided in the Supplementary Material. For the first hidden layer, there is no preceeding layer activity and is instead the given task input $\mathbf{s}^{task}(t)$. Importantly, when estimating the forward pass of the network through the digital twins this same equation is used but for $\tilde{\mathbf{z}}_l^\pi(t)$ resulting in $\tilde{\mathbf{S}}_l(t)$ as the simulated input.

The output of the simulated network can be defined as a linear combination of the final layer activities, $\mathbf{o}(t) = \boldsymbol{\theta}_{L+1}\mathbf{z}_L^\pi(t)$, and, thus, optimization can follow the minimization of

$$\mathcal{L} = \sum_{t=t_0} E\left(\tilde{\mathbf{o}}(t), \mathbf{o}^{task}(t)\right) \tag{5}$$

where $E(\cdot)$ is an appropriately chosen loss function operating on the simulated network output $\tilde{\mathbf{o}}(t)$ and target task output $\mathbf{o}^{task}(t)$.

Thus, in a forward pass of the simulated network, we integrate the stochastic differential equations (3) through time and calculate the output $\tilde{\mathbf{o}}(t)$. We highlight that in a feedforward structure, even with dynamical node, each can be ran layer by layer. That is, we can simulate the dynamics $\mathbf{z}_l(t)$, $\forall\, t$ before passing to layer $l + 1$. This process is outlined in the pseudo-code presented in Algorithm 1.

In the backward pass, the connectivity parameters $\boldsymbol{\theta}$ are trained via backpropagation through time (BPTT) on the simulated network. The task-dependent optimal parameters are extracted like-for-like for use in experiments, where the resulting connectivity is validated on physically defined devices. The supplementary information provides more details on the use of BPTT and truncated-BPTT for the simulated system.

This optimization is performed using PyTorch's automatic differentiation and the Adam optimizer. In both the MNIST and neuroprosthetics tasks, we were able apply BPTT across the entire sequence. For the Mackey-Glass regression task, which involves forecasting a sequence of arbitrary-length, we employed truncated BPTT, training on temporal segments with starting points randomly sampled from the dataset. Additionally, a burn-in phase was used at the start of each segment, during which the system was evolved without gradient tracking. This allows the dynamic system to settle into a representative regime before the learning process.

Performing this optimization process assumes that generated system responses, $\tilde{\mathbf{y}}(t)$, are approximately equivalent to physical device activities, $\mathbf{y}(t)$, and that the unknown devices dependencies $\frac{d\mathbf{y}_i(t)}{d\mathbf{s}_i(t')}$ can be approximated through $\frac{d\tilde{\mathbf{y}}_i(t)}{d\tilde{\mathbf{s}}_i(t')}$ $\forall t$ and $t' < t$ in the temporal interval considered. BPTT, or truncated BPTT, will then decompose such total derivatives into the terms $\frac{\partial \tilde{\mathbf{y}}_i(t')}{\partial \tilde{\mathbf{s}}_i(t'-\delta t)}$ and $\frac{\partial \tilde{\mathbf{y}}_i(t')}{\partial \tilde{\mathbf{y}}_i(t'-\delta t)}$ $\forall t'$ in the considered interval.

Although our discussion centers on an MLP architecture, the approach generalizes naturally to any feedforward network. The only modification required is to replace the linear transformation in Eq. (4) with the appropriate operation for the chosen architecture.

## Nanomagnetic Ring Arrays (NRA)

**Fabrication of ring arrays.** Wafers of Si (001) with a thermally oxidized surface were spin-coated with 200nm of positive resist, with the nanoring array geometries and electrical contacts patterned via electron-beam lithography using a RAITH Voyager system. The magnetic nanoring arrays were patterned, then metallized to nominal thicknesses of 10nm via thermal evaporation of $Ni_{80}Fe_{20}$ powder using a custom-built (Wordentec Ltd) evaporator (typical base pressures of below $10^{-7}$ mBar), before removal of the initial resist. Electrical contacts were patterned via a second lithography stage and were metallized via two-stage thermal evaporation of a 20nm Ti seed layer followed by a 100nm layer of Au.

**Electrical transport measurements of ring arrays.** Rotating magnetic fields were generated at 64 Hz via two pairs of air-coil electromagnets each with a voltage-controlled Kepco BOP 36-6D power supply. A sinusoidal voltage wave of 13,523 Hz was generated via an Aim-TTI instruments TG1000 signal generator and an SRS C5580 current source to generate 2 mA current, which was then injected to the nanoring arrays via the electrical contact pads. A National Instruments NI DAQ card measured the resulting potential difference across the device (modulated via anisotropic magnetoresistance (AMR) effects), sampling at 2 MHz. Lock-in amplification was performed digitally by multiplying the measured voltage signal with a digitally generated reference wave matching the input current frequency, before filtering via a digital low-pass filter with a cut-off frequency of 320 Hz to remove the kHz component and leave the AMR dependent signal. The filtered waveform was then downsampled to a rate of 3.2kHz (50 samples per rotation of applied field) to reduce data size. Further images of the

experimental setup and an overview of the signal path can be found in Supplementary Fig. 14.

**Neural-SDE models of NRAs.** Training data for the deterministic component of the neural-SDE (parameterized by **f**) was generated by driving the NRAs under 20,000 randomly generated sequences of 20 inputs, with the applied rotating magnetic fields spanning the responsive range of the devices, and recording the resulting AMR signals. The external signal $s(t)$ represented the magnitude of the applied field at time $t$. Five delays were used here to define the hidden state, $\mathbf{y}(t) = (x(t), \ldots, x(t-5))$, where the $x(t)$ represented the measured AMR signal. The data used to generated the stochastic component (parameterized by **g**) was generated by driving the system with 1000 randomly generated input sequences of length 20 for 100 repetitions to generate example distributions of the noisy measurements, with 10 auxilliary variables used to generate different timescales of noise.

## Artificial Spin Vortex Ices (ASVIs)
Part of the description of the experimental methodologies for the ASVIs is reproduced from earlier works of several of the authors[14].

**Fabrication of artificial spin vortex ices.** Artificial spin-ice arrays were fabricated via electron-beam lithography liftoff method on a Raith eLine system with PMMA resist. 25 nm $Ni_{81}Fe_{19}$ (permalloy) was thermally evaporated and capped with 5 nm $Al_2O_3$. The flip-chip FMR measurements require mm-scale nanostructure arrays. Each sample has dimensions of roughly ~ $3 \times 2$ mm. As such, the distribution of nanofabrication imperfections termed 'quenched disorder' is of greater magnitude here than typically observed in studies on smaller artificial spin systems, typically employing 10–100 micron-scale arrays. The chief consequence of this is that the Gaussian spread of coercive fields is over a few mT for each bar subset. Smaller artificial spin reservoir arrays have narrower coercive field distributions, with the only consequence being that optimal applied field ranges for reservoir computation input will be scaled across a corresponding narrower field range, not an issue for typical 0.1 mT or better field resolution of modern magnet systems.

**Spectral fingerprinting of artificial spin-vortex ices.** Ferromagnetic resonance spectra were measured using a NanOsc Instruments CryoFMR in a Quantum Design Physical Properties Measurement System. Broadband FMR measurements were carried out on large area samples (~$3 \times 2$ mm$^2$) mounted flip-chip style on a coplanar waveguide. The waveguide was connected to a microwave generator, coupling RF magnetic fields to the sample. The output from waveguide was rectified using an RF-diode detector. Measurements were done in fixed in-plane field while the RF frequency was swept in 10 MHz steps. The DC field was then modulated at 490 Hz with a 0.48 mT RMS field and the diode voltage response measured via lock-in. The experimental spectra show the derivative output of the microwave signal as a function of field and frequency[14].

**Neural-SDE models of ASVIs.** Training data for the deterministic component of the neural-SDE was gathered for a sequence of 13,000 inputs, with saturation pulses provided sporadically to reset the device. Data for the stochastic component was generated via 100 repetitions of 100 different input sequences of length 20. The ASVI response $\mathbf{x}(t)$ corresponds to the measured FMR spectra driven by an external field of amplitude $s(t)$. The dimensionality of $\mathbf{x}$ was sufficient to capture the system's dynamics and augmentation of the N-DE input variables was not necessary, setting $\mathbf{y}(t) = \mathbf{y}^\pi(t) = \mathbf{x}(t)$.

## Physical Neural Networks
**Physical systems as dynamic nodes.** The NRA devices were initialized via a single rotation of magnetic field at 80 Oe, with a sample from the distribution of the final AMR states of the initialization procedure used as initial conditions for the neural-SDE model. The ASVI devices were initialized via a linearly applied field of 235 Oe, with similar selection of initial conditions for the model. The feed-forward networks were constructed by repeated measurements of a single physical device of each class (one NRA device, one ASVI), mimicking the flow of information through the network by serially sampling the same device. Input data were combined with the transferred weights, then encoded into the strength of the applied magnetic fields and provided to the devices node-by-node within a given layer. The outputs of each layer were then combined with their respective weights and passed to the next layer in the network where the process was repeated. This hybrid between digitally stored network weights and physical nodes is due to current experimental limitations rather than limitations of the framework.

**Partially Observable MNIST.** The data for this task converts the original 784 dimensional input of the MNIST digits into a sequence of length N with 784 input features per step. At each timestep, the information from 784/N pixels is given via random sampling, and removed from the sample pool for subsequent images in the sequence. Hence, all of the information from the digit is provided by the end of the sequence. There is no correlation between the sampling process across multiple digits, resembling different permutations of information for every digit. Classification occurs from activities at the end of the sequence only. Training was performed via backpropagation through time in simulation, with hyperparameters tuned against a small validation set also in simulation. Testing was performed on networks of real devices on 1000 samples of unseen data, with reported accuracies averaged over three experimental runs with different masking of data.

**Movement classification of a neuroprosthetic device.** For the neuroprosthetic task, we adopted the second classification task (exercise B) from the Ninapro database[45], where sEMG activities have been recorded for 27 subjects. In these settings, the physical neural network is asked to perform gesture recognition from the sEMG recordings for all subjects. The sEMG temporal data has been preprocessed through a low-pass filter and sub-sampled at 100 Hz, leading to input sequences of 30 time steps for each gesture. The data was split into training, validation and testing sets, where the validation set was used to tune the hyperparameters. For optimization, we adopted truncated back-propagation through time to reduce simulation time and memory requirements. Particularly, we fixed the number of temporal steps over which dependencies are considered and BPTT is carried out to ten. The seventeen classes were represented via one-hot encoding of output neurons, with the target signal of the same length as the input data. Training was performed by optimizing the model output over a reduced window of the original signal, corresponding to the most meaningful information on the gesture (timestep 12 to 18). This meant that parts of the signal which are not informative for classification do not disrupt the learning process. This produced a single model which was able to perform decently over many time steps of prediction, shown in Supplementary Fig. 13.

**Mackey–Glass future prediction and cascade learning.** The signals used for prediction were generated via the following delay-differential equations[17,51]: $\frac{dx}{dt} = \frac{\alpha x(t-\tau)}{1 + x(t-\tau)^n} - \beta x(t)$, with $\alpha = 0.2$, $\beta = 0.1$, $\tau = 17$, $n = 10$, and $x_0 = 1.2$, solved numerically with a fourth-order Runge–Kutta solver and a timestep $dt = 2$, producing quasi-periodic behavior on the order of 25 samples. 5100 samples were generated, with the first 100 samples discarded as a wash-out of initial conditions. The next 1000 samples were used for training, and the same 1000 samples were used to gather the corrective datasets for the cascade learning approach. Model performance was evaluated in experiments over the

remaining 4000 samples in 10 subsections of 400 points, with the resulting error bars reflecting performance over the 10 sections.

## Data availability

The processed data presented figures in this manuscript can be found on ORDA at https://orda.shef.ac.uk/articles/dataset/Research_data_for_Noise-Aware_Training_of_Neuromorphic_Dynamic_Device_Networks_/29835680, under the following https://doi.org/10.15131/shef.data.29835680. For raw experimental data, contact i.vidamour@sheffield.ac.uk.

## Code availability

The code used in this manuscript can be accessed at https://github.com/LucaManneschi/NoiseAwareTwins_Project.

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

## Acknowledgements

L.M., I.V., T.J.H., M.O.E., and E.V. acknowledge funding from the EPSRC MARCH Project No. EP/V006339/1. M.O.E., T.J.H., and E.V. also acknowledge funding from the EPSRC Project No. EP/S009647/1. D.G. and S.S. acknowledge funding from the EPSRC MARCH Project No. EP/V006029/1. K.D.S. was supported by Schmidt Sciences, LLC. and the Engineering and Physical Sciences Research Council (Grant no. EP/W524335/1). G.V. and T.J.H. were funded via Horizon 2020 FET-Open SpinEngine (Agreement No 861618). L.M. and E.D. acknowledge support via the Royal Society through International Exchanges Award IEC\NSFC\22343. J.C.G. was supported by the Royal Academy of Engineering under the Research Fellowship program, the EPSRC ECR International Collaboration Grant 'Three-Dimensional Multilayer nanomagnetic Arrays for Neuromorphic Low-Energy Magnonic Processing' EP/Y003276/1, by ERC Starting Grant MORPHON, and by Imperial College London President's Excellence Fund for Frontier Research. W.R.B., J.C.G., and K.D.S. were supported by EPSRC grant EP/X015661/1.

## Author contributions

L.M. and M.O.E. developed the Neural-SDE framework. L.M., I.T.V. and K.D.S. performed all model fitting to experimental devices. I.T.V. performed experimental measurements of nanorings. K.D.S. performed experimental measurements of spin-ices. C.S. and G.V. fabricated nanoring samples. K.D.S. and J.C.G. fabricated spin-ice samples. I.T.V., C.S. and G.V. developed the experimental setup for nanoring measurement. D.G. optimized Neural-SDE code. K.D.S., L.G., D.S. and D.D. developed preliminary LSTM models for spin-ices. D.H. performed preprocessing of neuroprosthetic data. E.D. supported the development and analysis of the temporal classification benchmark, particularly for the electromyography component, and contributed to the revision of the entire manuscript. L.M., I.T.V., M.O.E., and E.V. drafted the manuscript. K.D.S., C.S., G.V., S.S. and J.C.G. provided primary reviewing and editing of manuscript. T.J.H. and W.R.B. provided feedback on experimental measurements for nanorings and spin-ices, respectively. E.V. provided technical input on model development throughout the project. All authors assisted in reviewing and editing of the manuscript. L.M. and E.V. conceived the work.

## Competing interests

The authors declare no competing interests.
