## [Transparent Peer Review file · Nature Communications]

Noise-Aware Training of Neuromorphic Dynamic Device Networks

Corresponding Author: Dr Ian Vidamour

Version 0:

Reviewer comments:

Reviewer #1

(Remarks to the Author)

This manuscript introduces an innovative noise-aware methodology for training networks of physical devices using Neural Stochastic Differential Equations (Neural-SDEs). These Neural-SDEs function as differentiable digital twins, capturing both the dynamics and stochastic behavior of devices with intrinsic memory. This contribution is significant in the field of physical computing, addressing the challenge of designing device networks for dynamic tasks in the absence of precise physical models. The proposed methodology shows potential for enhancing the robustness and scalability of physical neural networks.

However, here are some points that require clarification:

1. The manuscript includes temporal classification experiments with both the MNIST dataset and a Neuroprosthetic task. While MNIST serves as a synthetically generated temporal dataset, its original task does not inherently require temporal information. In contrast, the Neuroprosthetic task offers a more meaningful demonstration of the method's capabilities. A more detailed presentation of the Neuroprosthetic task results would significantly strengthen the evaluation of the proposed approach.
2. Several existing methods for training physical neural networks (PNNs), such as Physics-Aware Training (PAT)¹ and Physical Local Learning (PhyLL)², have demonstrated effectiveness. Including a comparative analysis with these methods is crucial to validate the advantages and effectiveness of the proposed Neural-SDEs.
3. The manuscript claims that the proposed approach can extend to arbitrarily deep dynamic PNNs. However, it would be helpful to clarify whether there is an upper limit on the number of layers supported. A discussion of any practical or theoretical constraints on depth would enhance the manuscript.
4. The manuscript introduces cascade learning as a solution to the training challenges of multi-layer physical neural networks, supported by experimental demonstrations. This strategy, which employs a greedy algorithmic approach, fixes the parameters of each layer after training for a regression task and trains subsequent layers based on the previously trained ones. While this method may yield locally optimal solutions, it does not guarantee global optimality. A comparative analysis with other joint-training approaches for multi-layer PNNs would provide greater insight into the effectiveness of cascade learning.
5. The paper experimentally demonstrates the applicability of Neural-SDEs to physical neural networks composed of nano-magnetic ring arrays (NRA) and artificial spin vortex ice (ASVI). However, it remains unclear whether this method represents a universal approach or if its applicability is limited to specific types of physical devices. A more detailed discussion on this point is needed for clarification.
6. In the temporal classification tasks presented in the manuscript, the baselines for the MNIST and Neuroprosthetic classification tasks differ. Notably, the baseline for the Neuroprosthetic classification task involves training the output layer,

which is not the case for the MNIST task. It would be better to elaborate on the rationale for employing these distinct baselines and provide additional details to help readers achieve a clearer understanding.

Experimental Details:

1. The manuscript lacks a description of the experimental system used for temporal classification and regression experiments. Including images and detailed descriptions of the system setup would improve clarity.
2. It is unclear whether the training data for the SDE is collected from a single device or multiple devices. Does the composed network in the classification and regression experiments correspond to multiple devices, or is a single device reused?
3. The manuscript states that Neural-SDEs achieve performance comparable to an MLP with a similar number of trainable parameters in the temporal classification task. More specific details, such as the network architecture, parameter count, and experimental accuracy, should be provided to substantiate this comparison.

Overall, the manuscript presents a novel approach, demonstrating its superiority in some aspects. However, additional comprehensive experiments and detailed discussions are necessary to fully validate the claims and strengthen the impact of the work.

[1] Wright, L.G., Onodera, T., Stein, M.M. et al. Deep physical neural networks trained with backpropagation. *Nature* 601, 549–555 (2022). <https://doi.org/10.1038/s41586-021-04223-6>

[2] Momeni, A., Rahmani, B., Malléjac, M., del Hougne, P. & Fleury, R. Backpropagation-free training of deep physical neural networks. *Science* 382, 1297–1303 (2023). DOI:10.1126/science.adi8474

(Remarks on code availability)

Reviewer #2

(Remarks to the Author)

Summary

The authors proposed a noise-aware training method for dynamical PNNs. Their approach is based on Neural Stochastic Differential Equations (Neural-SDEs), which incorporate realistic device-specific dynamics and noise. This is essential for closing the gap between noisy analog hardware and digital simulations. By modeling colored noise through auxiliary variables and performing (truncated) backpropagation through time (BPTT), the authors claim improved transfer of parameters from simulation to hardware.

Main Comment:

I agree that prior training methods, such as Physics Aware Training (PAT) and Physical Local Learning (PhyLL), primarily targeted static devices. However, I strongly believe they could support dynamic devices by employing recurrent neural networks (e.g., LSTMs) or neural ODEs as digital twins to estimate gradients. It should also be noted that both PAT and PhyLL rely on using the actual physical system during the forward pass, thereby inherently capturing system noise and narrowing the reality-simulation gap. Consequently, I can only recommend the paper for publication if the authors compare their results against at least the PAT method in a dynamic setting. For example, they could set up a scenario where (a) the actual physical device is directly used in the forward pass and a digital neural ODE twin in the backward pass to estimate the gradient (akin to PAT), versus (b) the authors' method that employs an in-silico noise-aware approach (Neural-SDE). Such a direct head-to-head comparison would significantly clarify the superiority of the proposed training scheme.

Other Comments

1- The authors use backpropagation through time (BPTT), which is well-known to have high memory overhead because it stores all intermediate states during the forward pass. This can make training on longer time horizons prohibitively expensive. To alleviate these issues, the authors could consider the adjoint method, which offers an $O(1)$ memory solution for ODEs and has extensions for SDEs. This approach can eliminate the need for truncated BPTT in many cases.

A relevant reference is:

Li, X., Wong, T. K. L., Chen, R. T., & Duvenaud, D. (2020). Scalable gradients for stochastic differential equations. In *International Conference on Artificial Intelligence and Statistics* (pp. 3870-3882). PMLR.

2- Please add more practical details (e.g., batch size, learning rate, size of datasets, number of training epochs) in a clear tabular form for each of the tasks and devices. I highly recommend releasing your dataset and source code or at least providing comprehensive pseudo-code in the supplementary material. Additionally, I had difficulty finding the number of trainable parameters for each PNN device.

3- Please benchmark PNNs performance vs. pure digital neural counterparts for the specified tasks

4- Please comment on the energy efficiency of the proposed PNN devices compared to standard digital hardware. While

discussing absolute energy numbers might be difficult, even approximate or relative comparisons can be illuminating, particularly given the motivation for energy-efficient neuromorphic computing.

5- Please discuss the feasibility and potential challenges of scaling the employed PNN devices.

6- Finally, please add a quantitative comparison between modeling with colored noise vs. a simpler i.i.d. white noise assumption. Providing a power spectral density (PSD) analysis or other metrics in the supplementary material could demonstrate the necessity of auxiliary variables in the Neural-SDE.

(Remarks on code availability)

Reviewer #3

(Remarks to the Author)

Manneschi et al. reported a study on a new method named noise-aware training to implement training strategy in physical neural networks (PNN). They developed a framework to training physical system that accounts for their dynamics and noise, extending beyond the previous Physics aware training (PAT) that can only train static physical devices. The authors have demonstrated the methodology and results on modified MNIST dataset and Mackey-Glass prediction tasks. PNN is an emergent field that can accelerate the training efficiency of artificial neural networks. The authors present an interesting idea that allows one to extend the PNN training towards more dynamical physical systems. However, several critical issues and missing results need to be well-addressed prior to the publication:

- 1) This work share similarity with the previous PAT paper [Nature 601, 549 (2022)] that one needs to build a digital twin model to simulate a real physical system. Such method has a critical issue known as “simulation-reality gap” since it is always difficult to accurately include all degrees of freedom and complex interactions in a real physical system by a model. Such issue causes performance degradation during both the training and inference stages. When extending to time-dependent dynamical memory data processing, one can imagine such inaccuracy will be significantly amplified. Especially, the authors seem to model the spin textures by only adapting the input-output experimental results without considering the real spin dynamics. Therefore, it is puzzling whether this framework can adapt such inaccuracy. I understand that characterizing noise can improve such issue as presented in the manuscript, but the main issue here is whether a numerical model can effectively simulate a real complex physical system when time is also involved.
- 2) The authors build two models, ODE and SDE to provide the deterministic and stochastic dynamics of the physical system, which is an interesting approach. However, this approach automatically decouples the correlation between the deterministic and stochastic (noise) signals since they are trained in separate networks. In real physical system, the level of noise signal is often correlated to the deterministic signal, either on its amplitude or time-dependent noise patterns. With such information missing, I doubt that one can guarantee the general performance of this training method.
- 3) The author lacks comprehensive comparison on the results to the standard digital training methods, for example, reservoir computing, spiking neural networks which is aimed to treat similar tasks. An overall comparison of the relevant benchmarks is important (more than Fig. S12), i.e., number of layers, size of weights and accuracy etc.

(Remarks on code availability)

Reviewer #4

(Remarks to the Author)

(Remarks on code availability)

Version 1:

Reviewer comments:

Reviewer #1

(Remarks to the Author)

The revision offers more comprehensive details regarding training networks of dynamic device using Neural-SDEs and has satisfactorily addressed the majority of my concerns. There remain some issues that should be addressed to support this paper's acceptance.

1. The manuscript mentions that the feed-forward networks were constructed by repeated measurements of a single physical device. However, in practical physical systems, the network is typically composed of multiple units. Since there often exist performance variations between different instances of the same physical device, it is important to discuss the consistency

across different units and how such variations might affect the proposed method.

2. For the Mackey-Glass signal prediction task, the revised manuscript provides a performance comparison with previous implementations. However, unlike the partially-observable MNIST and neuroprosthetic movement classification tasks, it does not include a comparison with digital neural network counterparts. Providing a digital-based baseline is essential for clearly demonstrating the advantages of the proposed method.

(Remarks on code availability)

The code is complete and fully covers the experiments described in the paper.

Reviewer #2

(Remarks to the Author)

The authors have provided complete and satisfactory answers to my comments. The revised manuscript is, in my opinion, of high quality and of interest to the readers of Nature Communications.

(Remarks on code availability)

I confess that I did not download the code to check it.

Reviewer #3

(Remarks to the Author)

The authors have addressed all my questions, and I am satisfied with the answers.

(Remarks on code availability)

Reviewer #4

(Remarks to the Author)

(Remarks on code availability)

Reviewer #1 (Remarks to the Author):

This manuscript introduces an innovative noise-aware methodology for training networks of physical devices using Neural Stochastic Differential Equations (Neural-SDEs). These Neural-SDEs function as differentiable digital twins, capturing both the dynamics and stochastic behavior of devices with intrinsic memory. This contribution is significant in the field of physical computing, addressing the challenge of designing device networks for dynamic tasks in the absence of precise physical models. The proposed methodology shows potential for enhancing the robustness and scalability of physical neural networks.

However, here are some points that require clarification:

The authors would like to thank the reviewer for their careful review, and their kind remarks on the innovation and the impact of the work. We hope the following amendments to the manuscript in light of their comments help improve the clarity and robustness of our submission.

1. The manuscript includes temporal classification experiments with both the MNIST dataset and a Neuroprosthetic task. While MNIST serves as a synthetically generated temporal dataset, its original task does not inherently require temporal information. In contrast, the Neuroprosthetic task offers a more meaningful demonstration of the method's capabilities. A more detailed presentation of the Neuroprosthetic task results would significantly strengthen the evaluation of the proposed approach.

We agree with the reviewer that the neuroprosthetic demonstration is the most impactful study. In light of this, we have expanded the discussion around the neuroprosthetics task section of the paper:

“To extend our study to a more challenging task, the NRA networks were trained to recognise hand gestures for controlling a neuroprosthetic device \cite{atzori2014electromyography}. This task used real-world electromyography (EMG) data collected from the forearms of patients performing seventeen different hand and wrist movements. Predictions were based on the class with the highest output within a window corresponding to data acquired between 120–180 ms after the gesture onset (see~\ref{fig:Supp_pNeuro}). The integration of neuromorphic systems with EMG data presents a promising avenue for addressing the challenges of real-time temporal classification~\cite{Donati_etal19, Vitale_etal22}, with the potential to leverage low-energy computation to improve the efficiency of gesture recognition for neuroprosthetic applications.”

And

“To demonstrate applicability to real-world problems, we applied the NADO approach to a neuroprosthetics task \cite{atzori2014electromyography}, where the goal was to classify human gestures using surface electromyography data from ten forearm electrodes. This task is substantially more challenging than the partially observable MNIST benchmark due to

longer input sequences accumulating more experimental noise, and the need for generalisation across both gestures and different subjects.”

2. Several existing methods for training physical neural networks (PNNs), such as Physics-Aware Training (PAT)¹ and Physical Local Learning (PhyLL)², have demonstrated effectiveness. Including a comparative analysis with these methods is crucial to validate the advantages and effectiveness of the proposed Neural-SDEs.

The authors thank the reviewer for raising this important point. While previous approaches have demonstrated effective training methodologies for static physical neural networks, they have not yet been applied to dynamic systems with physical memory, such as those studied here. To enable a direct comparison with our methodology, we extended PAT—the approach most closely aligned with ours—to dynamic devices, as suggested by Reviewer 2. A detailed description of this extension and the corresponding results is provided in the supplementary material, where we show that a stochastic model is necessary to capture bifurcations and appropriately guide parameter learning within the gradient calculations (see Figure 14 and our response to Reviewer 2).

Our methodology is not intended to fully replace existing approaches such as PAT and PhyLL, but rather to serve as a complementary tool that augments these frameworks. We have expanded the Discussion section to clarify this point.

Figure 13 - Comparison of training times for a dynamic physical neural network (PNN) using the Neural-ODE methodology with PATs, versus training on the ODE alone, as a function of sampling speed for experimental devices. Vertical lines indicate the sampling rates achievable by ASVIs (2 Hz) and NRAs (64 Hz). Quoted times correspond to 30,000 iterations with a batch size of 50 and a signal length of 10 inputs, for a network comprising 200 nodes—requiring a total of 3 billion samples. The blue line is calculated by multiplying the number of samples by the acquisition time, accounting also for computation time. As simulation is independent of experimental throughput (though the computational time is dependent on model complexity; time estimates here are for an SDE model of NRA devices,

and total simulation time stands at just under 8 hours for network optimisation), it represents the lower bound on training duration in both methodologies.

Extending the PAT framework to dynamic settings requires correcting device trajectories, which in turn necessitates multiple measurements. As a result, PAT becomes challenging to apply in scenarios where experimental measurements are time-consuming, as in our work. This is illustrated in Supplementary Figure 18, where we compare estimated runtimes for PAT + NeuralSDE versus NeuralSDE alone. Training time increases substantially unless the data acquisition rate is much higher than in the systems we employed.

Due to these experimental constraints that made direct physical measurements impractical, we performed an alternative comparison entirely in simulation. We used analytical models of noisy leaky integrators and Duffing oscillators as device models. Training was conducted using our framework with neural SDEs and an extended PAT framework, employing neural ODEs as digital twins. We selected ODEs in accordance with Reviewer 2's suggestion, noting that PAT uses noise-free models for gradient calculation (backward pass) while relying on noisy experimental measurements for output computation (forward pass). Our results demonstrate that, without incorporating noise—especially information about bifurcation—into the backward pass (gradients), optimization does not achieve the same level of performance as when this knowledge is included.

In Discussion:

“Previous methods for training physical networks, such as Physics-Aware Training (PAT) \cite{wright_deep_2022} and Physical Local Learning (PhyLL) \cite{momeni2023_PhyLL}, have been limited to static devices. In contrast, our framework embraces the dynamical nature of physical systems, treating this complexity not as a hindrance but as a computational asset. To our knowledge, this is the first demonstration that interconnected physical devices can be optimized using backpropagation through time (BPTT), the foundational learning algorithm for recurrent neural networks and dynamical systems.

Our use of digital twins shares conceptual parallels with PAT but extends to dynamically driven, noise-aware models. Whereas PAT relies on experimental measurements to correct a model's internal activity and reduce the simulation–reality gap during training, our neural SDE framework enables effective optimization without requiring real-world data for optimisation in classification tasks.

Extending PAT to the dynamical setting involves correcting the system's computational graph through time (see Supplementary Material – Generalizing PAT to Dynamical Settings). This requires sampling device states across temporal trajectories and adjusting estimates of both system state evolution and input–output dependencies. In the supplementary material, we detail this generalization and evaluate performance as a function of the adopted sampling strategy. Notably, sampling every device at every time step for each input signal during training is experimentally demanding, and practically infeasible with the hardware considered here, due to relatively slow experimental throughput. This limitation is compounded by the inability to parallelize across batch sizes in hardware.

Nevertheless, PAT-inspired sampling strategies remain valuable for refining model behavior. Our neural SDE framework should not be seen as a replacement, but as a complementary alternative. There is no intrinsic barrier to applying experimental corrections to neural SDE models to further reduce this gap. For the regression task investigated, we employed a cascade learning approach—interpretable as a sparse variant of PAT—to incrementally correct neural SDE activity layer by layer. This enabled us to balance theoretical performance with experimental feasibility. As observed in regression tasks, some degree of sampling was necessary; however, satisfactory performance was achieved without continuous correction through time or at every parameter update.

Adopting neural SDEs as the underlying physical model provides a robust mechanism for generating noisy samples during digital training and yields a differentiable representation of device stochasticity. This enables gradients to be backpropagated through the stochastic component itself—a critical feature for systems whose responses depend non-trivially on specific noise realizations, which PAT alone cannot account for (see \ref{fig:Supp_perfBaseline_PAT}).

As the extension of the PAT framework to dynamical settings—and its associated methodology—is extensive, we refer the reviewers to the main manuscript for details, in order to maintain brevity here and ensure correct formatting of mathematical expressions.

These changes are summarized in the supplementary material under “Generalizing PAT to dynamical settings” and “Network forward pass.” Additional implementation details on constructing the computational graph are provided in “BPTT and the computational graph,” with further comparisons discussed in “Comparison with the PAT framework.”

3. The manuscript claims that the proposed approach can extend to arbitrarily deep dynamic PPNs. However, it would be helpful to clarify whether there is an upper limit on the number of layers supported. A discussion of any practical or theoretical constraints on depth would enhance the manuscript.

We would like to thank the reviewer for highlighting this lack of clarity. Because of building the network layer by layer, and always sample training data for the the penultimate hidden layer from the device, the error due to the model-device error is bounded. This is detailed in the newly added Supplementary Material section, “Cascade learning and isolating error propagation”. As shown in Figure 4 (main), cascade learning improves the performance of transferred networks. However, the constraints it imposes on the learning algorithm do not guarantee performance equivalent to full optimization in the absence of simulation–reality mismatch (as discussed in the next response; see also Figure 12 in the Supplementary Material).

Practically, cascade learning does not necessarily improve performance with the addition of further layers—though this limitation is also well-known in standard neural networks. A comparative analysis between cascade learning and full backpropagation is provided in the following response.

4. The manuscript introduces cascade learning as a solution to the training challenges of multi-layer physical neural networks, supported by experimental demonstrations. This strategy, which employs a greedy algorithmic approach, fixes the parameters of each layer after training for a regression task and trains subsequent layers based on the previously trained ones. While this method may yield locally optimal solutions, it does not guarantee global optimality. A comparative analysis with other joint-training approaches for multi-layer PNNs would provide greater insight into the effectiveness of cascade learning.

While no gradient-based technique can guarantee global optimality, we agree that the constraints imposed by layer-wise training can influence the final accuracy of a given model.

For clarity, we interpret “global optimality” here as the performance achievable by jointly optimizing all network weights simultaneously, as in standard backpropagation. Although our experimental results show that cascade learning improves the transferability of Neural SDE-optimized networks (Figure 5), we agree that a direct comparison with joint-training approaches for multi-layer physical neural networks would provide further insight.

To address this, we conducted additional simulations using networks of analytically defined, noiseless leaky integrator neurons, where joint optimization via standard backpropagation through time was tractable. The results (Figure 12) show that, for our tasks, the performance difference between cascade learning and joint training is not significant—indicating that cascade learning remains a practical and effective strategy for physical systems. These results and further discussion have been included in the revised manuscript.

The following discussion has been added to the supplementary material:

“Empirical limitations of cascade learning on performance.

To compare the performance of cascade learning with fully optimized networks unaffected by simulation–reality mismatch, we repeated the Mackey–Glass time series prediction task using simulations of analytically defined leaky integrator neurons, forming a dynamical feedforward network consistent with our study. Figure \ref{fig:Cascade} presents results for predicting the first five future time steps of the input sequences used in the main text, evaluated across increasing numbers of hidden layers. For both approaches, 40,000 training iterations were performed on sequences of length 90, with backpropagation through time applied only to the final 60 inputs, and a batch size of 50 sequences.

We observe that the cascade learning approach tends to saturate earlier with respect to depth, and generally converges to slightly lower performance than full optimization. While cascade learning improved performance in our physical experiments—by isolating error propagation due to simulation–reality mismatch—in the idealised case of noiseless, perfectly modelled systems, full optimization yields superior results.”

An additional comment has also been added to the main text:

“However, as in any machine learning network, an improvement in performance is not guaranteed by adding additional hidden layers, and the configurations learned via cascade learning may be sub-optimal when compared to full optimisation in the absence of simulation-reality mismatch, with techniques such as PAT\cite{wright2022deep} serving as

useful methodologies for minimising this gap where experimental throughput allows reasonable training times (see Supplementary Figures \ref{fig:Cascade} and \ref{fig:Training_Time}).”

5. The paper experimentally demonstrates the applicability of Neural-SDEs to physical neural networks composed of nano-magnetic ring arrays (NRA) and artificial spin vortex ice (ASVI). However, it remains unclear whether this method represents a universal approach or if its applicability is limited to specific types of physical devices. A more detailed discussion on this point is needed for clarification.

The authors agree that the universality of the method is crucial in assessing its general relevance. Neural-SDEs can, in principle, be applied to any dynamical system, including those that are only partially observable, provided that the state space is appropriately augmented to account for unmeasured variables, as is standard practice in machine learning. This assumption underlies the generality of our approach.

While we only have partial observability of the system state through the output, we augment this information using a buffer of previous output states to approximate the latent states. We demonstrate that this approach is also applicable to analytically defined dynamical systems such as leaky integrators and Duffing oscillators (Figures 5, 7, and 10) in addition to the experimental devices (Figures 2, 11, and 19). These latent states provide insight into how the system evolves dynamically, enabling the network to infer rates of change. Owing to the nature of the stochastic differential equations–based approach, we believe that any system whose evolution can be described by a set of underlying differential equations can, in principle, be modeled via our paradigm.

Practically, this requires providing the neural network with sufficient information to capture higher-order differential terms necessary for accurate numerical integration. Previous work has shown that this is possible by using a set of delayed output states as inputs to the neural network (see Chen et al., Nature Communications 13, 2022, “Forecasting the outcome of spintronic experiments with Neural Ordinary Differential Equations”), a technique that we have also employed here. Provided that a sufficient number of delayed measurements of the system state are available to approximate Markovian dynamics, the modeling technique should be applicable.

To highlight this point in the main text, the following discussion has been added:

“Figure \ref{fig:Fig2_scheme}(c) shows the architecture of the proposed Neural-SDE model, comprising two neural networks: one for deterministic dynamics and another for stochastic dynamics. These networks are coupled via the numerical integration method, enabling the model to represent explicitly how system output depends on device state and external inputs, incorporating both deterministic and stochastic components. This structure supports noise-aware gradient computation for backpropagation through time (BPTT). The Neural-SDE architecture thus parameterizes the stochastic differential equations that define how device output evolves as a function of the current state and external input.”

The deterministic network (upper) and the stochastic network (lower) each receive external input signals and a sequence of past device states. The history length must be sufficient to approximate the system as Markovian, ensuring that future states can be predicted from current inputs and recent device states. When this criterion is met, the method is applicable to any dynamical system. The stochastic network also receives auxiliary variables to support noise modeling. Outputs from both networks are integrated using a stochastic numerical scheme to produce the device's activity (readout) at the next timestep. This value is recursively fed back as the most recent device state. Orange arrows in Figure \ref{fig:Fig2_scheme}(c) indicate error gradients with respect to device activity and external input, as computed during BPTT.

6. In the temporal classification tasks presented in the manuscript, the baselines for the MNIST and Neuroprosthetic classification tasks differ. Notably, the baseline for the Neuroprosthetic classification task involves training the output layer, which is not the case for the MNIST task. It would be better to elaborate on the rationale for employing these distinct baselines and provide additional details to help readers achieve a clearer understanding.

We would like to thank the reviewer for highlighting this lack of clarity. We would like to highlight that the output layers are trained in both baseline cases. To highlight the need for optimised connectivity between hidden layers, it is shown in the MNIST task that having multiple layers connected randomly—even if sampled from the same distribution as optimised weights—causes performance to suffer due to the inability of capturing meaningful relationships between hidden layers. The following text has been added to clarify this point:

“The baseline performance, shown as red dashed lines, corresponds to physical neural networks with the same architecture but randomised hidden-layer connectivity, following the reservoir computing paradigm for both one-layer (left, orange) and two-layer (right, green) networks. The hidden weights are randomly drawn from distributions matched to those of the Neural-SDE-optimised network (see \ref{fig:Supp_perfBaseline_Rings}b), while only the output weights are optimised. This demonstrates that achieving high performance requires optimisation of the entire network connectivity, not just the output layer.”

And

*“Figure \ref{fig:Fig3}(d) shows the measured performance of a network whose connectivity was optimized in simulation using digital twins and then transferred to the physical system. As a baseline, we used a randomly connected network with a single hidden layer and trained only the output weights, following the standard reservoir computing (RC) approach, with the number of nodes matched to the PNN. This adjustment from previous baselines avoids the performance degradation seen with random connections in deeper networks while maintaining the same node count. The optimized network achieves an error rate approximately 30% lower than the baseline, demonstrating the advantages of optimized connectivity over conventional *in-materio* RC and highlighting the improved capability made possible by connectivity optimization.”*

Experimental Details:

1. The manuscript lacks a description of the experimental system used for temporal classification and regression experiments. Including images and detailed descriptions of the system setup would improve clarity.

Additional figures have been added to the supplementary material to show the experimental system in more detail (Fig 16 for the NRA measurements), and details of the experimental setups are provided in Methods:

Figure 18. a Scanning electron micrograph of nanoring array, showing electrical contacts patterned on top of the nanoring structure. b Image of the sample mounted between electromagnet coils. c Schematic diagram for the experimental measurement of magnetic nanoring arrays. A desktop PC (green) interfaces between the analogue input and output functionality of a National Instruments data acquisition card (orange), which sends output voltage signals to a pair of voltage-controlled power supply units, which in turn provide rotating magnetic fields via two pairs of electromagnets (grey) uniformly over the magnetic nanoring array (black). A signal generator provides a uniform oscillating current signal to both the nanoring array, where the signal is modulated via anisotropic magnetoresistance effects, and also to a lock-in amplifier as a reference signal to filter out experimental noise. The filtered signal from the lock-in amplifier is passed to the analogue input of the data acquisition card which is then logged on the PC.

Descriptions of the experimental systems can be found in the Methods section:

“Fabrication of Ring Arrays

Wafers of Si (001) with a thermally oxidised surface were spin-coated with 200 nm of positive resist, with the nanoring array geometries and electrical contacts patterned via electron-beam lithography using a RAITH Voyager system. The magnetic nanoring arrays were patterned, then metallised to nominal thicknesses of 10 nm via thermal evaporation of Ni₈₀Fe₂₀ powder using a custom-built (Wordentec Ltd) evaporator (typical base pressures of below 10⁻⁷ mBar), before removal of the initial resist. Electrical contacts

were patterned via a second lithography stage and were metallised via two-stage thermal evaporation of a 20 nm Ti seed layer followed by a 100 nm layer of Au.

Electrical Transport Measurements of Ring Arrays

Rotating magnetic fields were generated at 64 Hz via two pairs of air-coil electromagnets each with a voltage-controlled Kepco BOP 36-6D power supply. A sinusoidal voltage wave of 13,523 Hz was generated via an Aim-TTI instruments TG1000 signal generator and an SRS C5580 current source to generate 2 mA current, which was then injected to the nanoring arrays via the electrical contact pads. A National Instruments NI DAQ card measured the resulting potential difference across the device (modulated via anisotropic magnetoresistance (AMR) effects), sampling at 2 MHz. Lock-in amplification was performed digitally by multiplying the measured voltage signal with a digitally generated reference wave matching the input current frequency, before filtering via a digital low-pass filter with a cut-off frequency of 320 Hz to remove the kHz component and leave the AMR dependent signal. The filtered waveform was then downsampled to a rate of 3.2kHz (50 samples per rotation of applied field) to reduce data size. *Further images of the experimental setup and an overview of the signal path can be found in Figure \ref{AMR_fig}.*

Fabrication of Artificial Spin Vortex Ices

Artificial spin reservoirs were fabricated via electron-beam lithography liftoff method on a Raith eLine system with PMMA resist. 25 nm Ni₈₁Fe₁₉ (permalloy) was thermally evaporated and capped with 5 nm Al₂O₃. The flip-chip FMR measurements require mm-scale nanostructure arrays. Each sample has dimensions of roughly ~ 3x2 mm. As such, the distribution of nanofabrication imperfections termed 'quenched disorder' is of greater magnitude here than typically observed in studies on smaller artificial spin systems, typically employing 10-100 micron-scale arrays. The chief consequence of this is that the Gaussian spread of coercive fields is over a few mT for each bar subset. Smaller artificial spin reservoir arrays have narrower coercive field distributions, with the only consequence being that optimal applied field ranges for reservoir computation input will be scaled across a corresponding narrower field range, not an issue for typical 0.1 mT or better field resolution of modern magnet systems.

Spectral Fingerprinting of Artificial Spin-vortex Ices

Ferromagnetic resonance spectra were measured using a NanOsc Instruments cryoFMR in a Quantum Design Physical Properties Measurement System. Broadband FMR measurements were carried out on large area samples (~ 3 × 2 mm²) mounted flip-chip style on a coplanar waveguide. The waveguide was connected to a microwave generator, coupling RF magnetic fields to the sample. The output from the waveguide was rectified using an RF-diode detector. Measurements were done in a fixed in-plane field while the RF frequency was swept in 10 MHz steps. The DC field was then modulated at 490 Hz with a 0.48 mT RMS field and the diode voltage response measured via lock-in. The experimental spectra show the derivative output of the microwave signal as a function of field and frequency."

2. It is unclear whether the training data for the SDE is collected from a single device or multiple devices. Does the composed network in the classification and regression experiments correspond to multiple devices, or is a single device reused?

The authors agree with the reviewer that this information should be clear and obvious in the manuscript. While repeated sampling of a single device is mentioned in the original manuscript, to highlight this point more clearly, the following text has been amended:

(In methods- sampling device behaviours)

“For each of the device classes, a single device was repeatedly sampled. First, the range of inputs at which the devices are dynamically active was established by sweeping input stimuli and observing changes in measured output. Data used for training the models of dynamic behaviours were sampled randomly from the determined input range...”

(In methods- Physical Neural Networks- Physical systems as dynamic nodes)

“The feed-forward networks were constructed by repeated measurements of a single physical device of each class (one NRA device, one ASVI), mimicking the flow of information through the network by serially sampling the same device. Input data were combined with the transferred weights, then encoded into the strength of the applied magnetic fields and provided to the devices node-by-node within a given layer. The outputs of each layer were then combined with their respective weights and passed to the next layer in the network where the process was repeated. This hybrid between digitally stored network weights and physical nodes is due to current experimental limitations rather than limitations of the framework.”

3. The manuscript states that Neural-SDEs achieve performance comparable to an MLP with a similar number of trainable parameters in the temporal classification task. More specific details, such as the network architecture, parameter count, and experimental accuracy, should be provided to substantiate this comparison.

To provide a better comparison, the following details have been added to the main text, with more quantitative comparison:

“Two-layer networks of nanorings optimised through interacting neural-SDEs demonstrate accurate transfer to devices and achieve performance exceeding that of static software-based MLPs with identical architectures (2 hidden layers of 200 nodes). These networks also match the performance of dynamic MLPs incorporating leaky integrators (transferred accuracies for NRAs: 96.8% at 25% visibility and 96.0% at 20% visibility, versus 90.0% and 88.0% for static MLPs, and 96.9% and 96.7% for leaky-integrator MLPs; see Supplementary Figure 10). These results show that the magnetic nanoring PNN effectively exploits the intrinsic device dynamics as a memory resource, underscoring the capability of the proposed framework to leverage physical dynamics for in-memory computing. However, the limited memory of NRAs results in a relative decline in performance at lower visibility compared to software leaky integrator networks.”

Overall, the manuscript presents a novel approach, demonstrating its superiority in some aspects. However, additional comprehensive experiments and detailed discussions are necessary to fully validate the claims and strengthen the impact of the work.

We would like to again thank the reviewer for their careful analysis of the manuscript, and hope that our amendments address their concerns fully and leave the manuscript suitable for publication.

- [1] Wright, L.G., Onodera, T., Stein, M.M. et al. Deep physical neural networks trained with backpropagation. *Nature* 601, 549–555 (2022). <https://doi.org/10.1038/>
- [2] Momeni, A., Rahmani, B., Malléjac, M., del Hougne, P. & Fleury, R. Backpropagation-free training of deep physical neural networks. *Science* 382,1297-1303(2023). DOI:10.1126/science.adi8474

Reviewer #2 (Remarks to the Author):

Summary

The authors proposed a noise-aware training method for dynamical PNNs. Their approach is based on Neural Stochastic Differential Equations (Neural-SDEs), which incorporate realistic device-specific dynamics and noise. This is essential for closing the gap between noisy analog hardware and digital simulations. By modeling colored noise through auxiliary variables and performing (truncated) backpropagation through time (BPTT), the authors claim improved transfer of parameters from simulation to hardware.

We would like to thank the reviewer for their careful insight and detailed analysis of our manuscript. We hope that the responses to their feedback addresses any outstanding concerns and improves our manuscript.

Main Comment:

I agree that prior training methods, such as Physics Aware Training (PAT) and Physical Local Learning (PhyLL), primarily targeted static devices. However, I strongly believe they could support dynamic devices by employing recurrent neural networks (e.g., LSTMs) or neural ODEs as digital twins to estimate gradients.

We thank the reviewer for highlighting this important point. We agree that prior methods such as Physics Aware Training (PAT) and Physical Local Learning (PhyLL) can, in principle, be extended to dynamic devices using recurrent neural networks (e.g., LSTMs) or neural ODEs as digital twins for gradient estimation. However, our central contribution is to demonstrate the necessity of noise-aware models for the backward pass in BPTT.

To address this, we have now extended PAT—the method conceptually closest to ours—to dynamic settings (Supplementary Material) and present two key findings: (1) trajectory correction requires multiple samples, which is prohibitive for slow-sampling devices such as ours, and (2) models that do not capture noise (e.g., neural ODEs) result in substantially reduced performance, especially for systems exhibiting phenomena like bifurcation. Supplementary Figure 13 now illustrates the experimental limitations of applying PAT to slow dynamical devices.

We also provide new experimental comparisons in Supplementary Figures 14 and 15. Figure 14 demonstrates the importance of model fidelity in the context of a Duffing oscillator, while Figure 15 shows that simple noise models are insufficient for robust performance. Additionally, Figure 4 in the main text highlights the superiority of Neural SDEs over Neural ODEs in our framework.

We further explored LSTM-based models as an alternative to neural ODEs. However, we found that standard LSTMs require significantly larger architectures and exhibit poorer performance (see Supplementary Figure 20). We hypothesise that explicit numerical integration schemes in Neural ODEs simplify the learning task by enabling direct parameterisation of partial gradients with respect to external inputs and device states. Our approach leverages the established framework of stochastic differential equations to incorporate noise explicitly.

To address these points, we have amended the main text in the section ‘Neural SDEs as differentiable digital twins’ and included the relevant supplementary figures comparing prediction accuracy between neural ODEs and LSTMs.

“In this work, we study two spintronic systems: nano-magnetic ring arrays (NRA)^{38-40,44} and artificial spin vortex ice (ASVI)^{3,21}. As illustrated in Figure~\ref{fig:Fig2_scheme}(b) for the NRA, device responses exhibit stochastic variation across repeated presentations of the same input sequence, with the response distribution shaped by both current and past inputs. This variability originates from intrinsic physical dynamics and experimental noise. To address the resulting simulation-reality gap, we extend the Neural stochastic differential equations (Neural SDE) framework to capture signal-dependent noise with complex autocorrelation. Compared to approaches such as long short-term memory networks (LSTMs), neural differential equations provide several advantages: higher predictive accuracy, implicit access to partial derivatives via numerical integration, and natural integration of stochastic dynamics through SDEs. Supplementary Figure~\ref{fig:ODE_LSTM} compares the prediction accuracy and number of trainable parameters for Neural ODE and LSTM models, and information on dataset construction and hyperparameter selection are tabulated in \ref{TrainingHyperparams}.”

Figure 20 - A comparison between achieved mean-squared error when using different modelling paradigms to model the response of the nanoring arrays and predict unseen input/output relationships for LSTM networks using the standard package within PyTorch, and the Neural-ODE approach. It can be seen that the ODEs vastly outperform LSTMs in

both terms of accuracy achieved, as well as parameter count. Training was performed across 50,000 iterations with a batch size of 50 and a signal length of 50 inputs, excluding the first 10 steps from backpropagation through time.

To evaluate the need for noise-aware optimization in the backward pass, we repeated the training process while adding noise to the forward pass through the Neural-ODE by evaluating the noise floor across data gathered for training the neural SDE, building a statistical model of how noise varies across inputs. This noise is present in the forward pass (as in PATs), but is not accounted for in the backward pass as this requires noise-aware differentiable models such as the framework we present. While this provides improved performance in transfer compared to the Neural-ODE alone, the model achieves worse performance compared to the neural SDE both in simulation and transfer, due to the inability of avoiding noisy regimes of device response as noise information is missing from the backward pass. The following results and text has been added to the manuscript:

We have extended the PAT framework to encompass dynamical settings (referring to manuscript to maintain brevity in this document):

Methods section under the ‘Generalising PAT to dynamical settings’, and also in ‘Network forward pass’. Further details on building appropriate computational graphs can be found under Supplementary material, ‘BPTT and the computational graph’, and further comparisons under ‘Comparison with the PAT framework’.

Additionally, the following has been added to the discussion:

Previous methods for training physical networks, such as Physics-Aware Training (PAT) \cite{wright_deep_2022} and Physical Local Learning (PhyLL) \cite{momeni2023_PhyLL}, have been limited to static devices. In contrast, our framework embraces the dynamical nature of physical systems, treating this complexity not as a hindrance but as a computational asset. To our knowledge, this is the first demonstration that interconnected physical devices can be optimized using backpropagation through time (BPTT), the foundational learning algorithm for recurrent neural networks and dynamical systems.

Our use of digital twins shares conceptual parallels with PAT but extends to dynamically driven, noise-aware models. Whereas PAT relies on experimental measurements to correct a model’s internal activity and reduce the simulation–reality gap during training, our neural SDE framework enables effective optimization without requiring real-world data for optimisation in classification tasks.

Extending PAT to the dynamical setting involves correcting the system’s computational graph through time (see Supplementary Material – Generalizing PAT to Dynamical Settings). This requires sampling device states across temporal trajectories and adjusting estimates of both system state evolution and input–output dependencies. In the supplementary material, we detail this generalization and evaluate performance as a function of the adopted sampling strategy. Notably, sampling every device at every time step for each input signal during training is experimentally demanding, and practically infeasible with the hardware considered here, due to relatively slow experimental throughput. This limitation is compounded by the inability to parallelize across batch sizes in hardware.

Nevertheless, PAT-inspired sampling strategies remain valuable for refining model behavior. Our neural SDE framework should not be seen as a replacement, but as a complementary alternative. There is no intrinsic barrier to applying experimental corrections to neural SDE models to further reduce this gap. For the regression task investigated, we employed a cascade learning approach—interpretable as a sparse variant of PAT—to incrementally correct neural SDE activity layer by layer. This enabled us to balance theoretical performance with experimental feasibility. As observed in regression tasks, some degree of sampling was necessary; however, satisfactory performance was achieved without continuous correction through time or at every parameter update.

Adopting neural SDEs as the underlying physical model provides a robust mechanism for generating noisy samples during digital training and yields a differentiable representation of device stochasticity. This enables gradients to be backpropagated through the stochastic component itself—a critical feature for systems whose responses depend non-trivially on specific noise realizations, which PAT alone cannot account for (see \ref{fig:Supp_perfBaseline_PAT}).

It should also be noted that both PAT and PhyLL rely on using the actual physical system during the forward pass, thereby inherently capturing system noise and narrowing the reality-simulation gap. Consequently, I can only recommend the paper for publication if the authors compare their results against at least the PAT method in a dynamic setting. For example, they could set up a scenario where

(a) the actual physical device is directly used in the forward pass and a digital neural ODE twin in the backward pass to estimate the gradient (akin to PAT), versus
(b) the authors' method that employs an in-silico noise-aware approach (Neural-SDE). Such a direct head-to-head comparison would significantly clarify the superiority of the proposed training scheme.

The authors agree that using the actual physical system for the forward pass, as in PAT and PhyLL, does indeed capture system noise and helps bridge the simulation–reality gap. However, the critical limitation of these previous methods is that they do not account for noise during the backward pass, i.e., during gradient estimation for training. This omission becomes especially consequential in dynamical systems, where system noise and bifurcations directly affect optimisation trajectories.

In the present work, we extend PAT to dynamic settings (see Supplementary, “Generalising PAT to dynamical settings”) and directly compare it to our noise-aware Neural-SDE approach. As shown in Supplementary Figure 14, we evaluate two analytical systems (the leaky integrator and the Duffing oscillator) by using the analytical model for the forward pass (mimicking access to real experimental data) and a neural ODE as the digital twin for the backward pass. We find that PAT can match or slightly outperform our method on the leaky integrator, but only if correction is applied at every step—requiring an impractical number of samples. In contrast, for the Duffing oscillator, PAT performs significantly worse, because noise and bifurcations are not modelled in the backward pass. In such cases, optimisation cannot avoid unstable or noisy regimes, and system performance degrades.

Figure 14- **a** Reference performance on the MNIST variation task (20% visibility) for two-layer networks based on analytical models. Neural-SDEs accurately capture system dynamics in both systems and yield optimised parameters robust to stochasticity-induced bifurcations, as seen in the Duffing oscillator. In contrast, neural-ODEs fail to model noisy system variants effectively. Applying the PAT framework to correct neural-ODE responses is successful for the stochastic leaky integrator, with performance strongly dependent on the sampling strategy used. However, PAT fails for the Duffing oscillator, where complex, noise-driven bifurcations occur. Without stochastic models for gradient estimation, optimisation cannot avoid regimes where bifurcation dominates. **b** Illustration of the impact of PAT noise correction frequency on MNIST task transfer accuracy. “Model/Transfer (end)” refers to a neural-ODE model and corresponding transfer accuracy for the leaky integrator where PAT correction is applied only at the final time step (“read-out,” when the classification decision is made). “Model/Transfer (10)” reflects performance when PAT correction is applied at every 10th integration step. The plot shows test accuracy over training time: large white circles (dashed lines) are neural-ODE model predictions, while smaller coloured dots (solid lines) are transfer results after applying trained parameters to the experimental system. “Model (10)” (high correction frequency) exhibits systematic but stable prediction error, indicating the digital twin remains effective for training. “Model (end)” (red curves) highlights pathological optimisation, where the model overfits deterministic digital features that have not been sufficiently corrected, resulting in declining performance as training proceeds. **c** Example of low-frequency PAT correction applied to a neural-ODE, showing the risk of using a deterministic model under sparse correction. The panel demonstrates that, after a brief period of variability at the correction point, the model reverts to a deterministic

trajectory, failing to capture ongoing stochastic behaviour. This result shows that for systems with fast or noise-driven dynamics, PAT correction must occur frequently enough to accurately simulate the true system behaviour.

Moreover, for physical systems like ours, the sampling requirements for PAT in the dynamical regime are prohibitive: as illustrated in Supplementary Figure 13, training on hardware for the MNIST task would take over a year due to the required number of repeated measurements, compared to hours in simulation. This makes direct head-to-head experiments infeasible with our devices, though the extended simulations provide a transparent comparison.

Figure 13 - Comparison of training times for a dynamic physical neural network (PNN) using the Neural-ODE methodology with PATs, versus training on the ODE alone, as a function of sampling speed for experimental devices. Vertical lines indicate the sampling rates achievable by ASVIs (2 Hz) and NRAs (64 Hz). Quoted times correspond to 30,000 iterations with a batch size of 50 and a signal length of 10 inputs, for a network comprising 200 nodes—requiring a total of 3 billion samples. The blue line is calculated by multiplying the number of samples by the acquisition time, accounting also for computation time. As simulation is independent of experimental throughput (though the computational time is dependent on model complexity; time estimates here are for an SDE model of NRA devices, and total simulation time stands at just under 8 hours for network optimisation), it represents the lower bound on training duration in both methodologies.

We further demonstrate (Supplementary Figure 15) that introducing simple statistical noise models—when these are absent from gradient calculations—still does not enable the network to learn how to avoid noisy regimes. As a result, both model prediction and transfer accuracy remain inferior to our noise-aware SDE approach (see also Supplementary Figure 12). Finally, we show that our Neural-SDE variant accurately models colored noise (Supplementary Figure 7).

In summary, although PAT and PhyLL capture noise in the forward pass, their inability to incorporate noise during gradient estimation (the backward pass) is a fundamental limitation for dynamical, noise-driven systems—one that our method directly addresses.

Other Comments

1- The authors use backpropagation through time (BPTT), which is well-known to have high memory overhead because it stores all intermediate states during the forward pass. This can make training on longer time horizons prohibitively expensive. To alleviate these issues, the authors could consider the adjoint method, which offers an $O(1)$ memory solution for ODEs and has extensions for SDEs. This approach can eliminate the need for truncated BPTT in many cases.

A relevant reference is:

Li, X., Wong, T. K. L., Chen, R. T., & Duvenaud, D. (2020). Scalable gradients for stochastic differential equations. In International Conference on Artificial Intelligence and Statistics (pp. 3870-3882). PMLR.

The authors agree upon the added value of discussing optimise then discretise vs discretise then optimise methods. Here, we chose the former due to its increased mathematical accuracy, and instead mitigated the increased memory overheads by truncating timesteps past beyond expected device memory (hence having infinitesimal influence on gradient calculation). Adjoint methodologies will become critical for devices with very long term memory, or when lower computational cost for optimisation is required.

The following text has been added to the manuscript:

(In Neural-SDEs as differentiable digital twins)

“When computing gradients through numerical integration schemes—such as those used for training models based on differential equations—two methodological classes are typically considered: optimize-then-discretize (indirect) and discretize-then-optimize (direct). In direct methods, like backpropagation through time (BPTT) applied to the integration scheme, the continuous system is first discretized, and differentiation is performed on this explicit sequence of operations. This approach provides mathematical precision by ensuring that gradients are computed exactly for the numerical scheme being used, but requires storing all intermediate states, resulting in memory costs that scale as $\mathcal{O}(t)$ with the length of the input signal t . In contrast, indirect methods such as the adjoint sensitivity method cite{pmlr-v108-li20i} take the continuous-time gradients first, then discretize, which allows gradients to be computed with constant $\mathcal{O}(1)$ memory cost with respect to signal length. However, this can sometimes introduce additional numerical error or instability, depending on the integration method and system dynamics.

In this work, we employ direct methods to maximise accuracy, as they most faithfully represent the discretization performed, and it matches the optimisation performed on the discriminator network in the GAN framework of the SDE (see Supplementary Figure 6c). To manage the associated memory demands, we truncate gradients beyond the intrinsic memory length of the device, since contributions from longer histories are negligible. For devices with much longer intrinsic memory, the resource demands of direct methods may be impractical, making indirect approaches more attractive despite their potential trade-offs.”

2- Please add more practical details (e.g., batch size, learning rate, size of datasets, number of training epochs) in a clear tabular form for each of the tasks and devices. I highly recommend releasing your dataset and source code or at least providing comprehensive pseudo-code in the supplementary material. Additionally, I had difficulty finding the number of trainable parameters for each PNN device.

The authors apologise that the above information was missing from the original submission. All of our code, annotated for readability, has been uploaded to a GitHub repository which can be freely browsed at https://github.com/LucaManneschi/NoiseAwareTwins_Project. Details of the training/testing iterations etc. have been added to supplementary figure 21:

ODE/SDE Training (Rings)	
Dataset (ODE)	20k patterns, length 20 (1800 train, 100 val + test)
Dataset (SDE)	2k patterns, 100 repetitions, length 20 (1800 train, 100 val + test)
Batch Size	200
Iterations	50k (ODE), then 50k (SDE)
Learning Rate	0.01
Time Horizon (BPTT)	50 discretisation steps (2.5 samples)

MNIST	
Dataset	80k images (60k train, 10k each val/test, 1k experimental transfer from test set)
Batch Size	50
Iterations	30k/70k (ODE), 70k (SDE)
Learning Rate	0.003
Time Horizon (BPTT)	Full signal (observability dependant)

Neuroprosthetics	
Dataset	27 patients, 17 gestures, 10 repetitions per gesture (8 train, 1 val, 1 test)
Batch Size	10
Iterations	50k
Learning Rate	0.003
Time Horizon (BPTT)	18 inputs (360 discretisation steps)

ODE/SDE Training (ASVI)	
Dataset	13k Samples (11k train, 1k each val/test)
Batch Size	50
Iterations	20k ODE, 10K SDE
Learning Rate	0.005
Time Horizon (BPTT)	10 Samples

Mackey-Glass (Cascade)	
Dataset	6000 samples (5k train, 500 val/test), First 1000 samples of train set repeated per additional layer for cascade reconstruction.
Iterations	5k (first ASVI Layer). 1 iteration of ridge regression for final output layer, with 5k iterations for updating input weights via cascade learning.
Batch Size	100
Learning Rate	0.001
Time Horizon (BPTT)	20 Samples

Figure 21: Details of hyperparameters selected for the optimisation and training data used for the neural-ODE/SDE models, partially-observable MNIST task, neuroprosthetic movement classification task, and Mackey-Glass future prediction task.

As we are learning the network connectivity only, the physical devices themselves have no trainable parameters. To make this point clearly in the main text, the following text has been added as a preface to the ‘Temporal Classification Benchmarks’ section of the main text:

“Here, we use the NADO framework to optimise the connectivity in networks of interacting devices. The physical devices represent each hidden node in the network and are treated as fixed nonlinear temporal kernels, with only the weights of the network optimised.”

3- Please benchmark PNNs performance vs. pure digital neural counterparts for the specified tasks

The authors agree that comparisons with digital architectures are important. While results for the MNIST task between digital neural networks with dynamic leaky integrator neurons are provided in supplementary material figure 10, there was no direct comparison, nor were they signposted in the main text. Additionally, the existing comparisons to software MLPs were qualitative and should instead be quantitative. To remediate this, the following text has been amended/added:

“Two-layer networks of nanorings optimised using interacting neural-SDEs demonstrate accurate transfer to physical devices and achieve performance exceeding that of static software-based MLPs with identical architectures (two hidden layers of 200 nodes). These networks also match the performance of dynamic MLPs incorporating leaky integrators (transferred accuracies for NRAs: 96.8% at 25% visibility and 96.0% at 20% visibility, versus 90.0% and 88.0% for static MLPs, and 96.9% and 96.7% for leaky-integrator MLPs; see Supplementary Figure 10). These results show that the magnetic nanoring PNN effectively exploits the intrinsic device dynamics as a memory resource, underscoring the capability of the proposed framework to leverage physical dynamics for in-memory computing. However, the limited memory of NRAs results in a relative decline in performance at lower visibility compared to software leaky integrator networks.”

Additional experiments were repeated for leaky-integrator based MLPs for the neuroprosthetics tasks (supplementary figure 14):

Figure 17: Experimental and reference performance on the neuroprosthetic task as the time at which the classification decision is made varies, for optimised networks of nanoring arrays (blue), analytical leaky-integrator based networks in simulation (orange), randomly connected networks of nanoring arrays (red), and standard software neural networks with small memory buffers of the previous three inputs (green). All networks are optimised to minimise cross-entropy loss between the network prediction and movement label over a

window containing the most meaningful information for classification, highlighted by the shaded orange region.

4- Please comment on the energy efficiency of the proposed PNN devices compared to standard digital hardware. While discussing absolute energy numbers might be difficult, even approximate or relative comparisons can be illuminating, particularly given the motivation for energy-efficient neuromorphic computing.

We would like to thank the reviewer for pointing out these important considerations. While we would like to stress the focus of the paper is upon the general methodology for training dynamic physical neural networks, to provide a better contextualisation, some calculations have been added to the supplementary material to show the current energy efficiency for writing and reading techniques that could be engineered for the relevant devices:

Energy Calculations for PNNs

As a baseline for the energy consumption for the dynamical neural networks in conventional hardware, we evaluate the number of floating point operations (FLOPs) per input for networks defined by analytical leaky integrators, as in the MNIST and neuroprosthetic tasks. The neuron update assumes the cheapest numerical integration method (Euler), meaning the effective equation for calculating neuron activity is defined by:

$$x_t = \tanh(\alpha * s_t) - (1-\alpha)x_{(t-1)}$$

where x_t describes activity at time t , s_t the input to the neuron at time t , and α the leak rate associated with the neuron. Depending upon the optimisation of the tanh operation, calculating this update equation costs between 6 and 25 FLOPs.

To calculate the cost per operation on both high-end GPU hardware (Nvidia RTX 4090), and typical computing platforms used for edge computation (Raspberry Pi 5), we take power consumption of each device (450 W for RTX 4090, 10 W for Raspberry Pi 5), and divide by the peak computational throughputs (82.6 TFLOPS for RTX4090, 31.4 GFLOPS for Raspberry Pi 5) to give an estimate for the energetic cost per floating point operation. This gives estimates for 4.45 pJ/FLOP for the RTX 4090, and 318 pJ/FLOP for the Raspberry Pi 5. Based on our earlier calculation for the number of FLOPs required per neuron update, this gives estimates for neuron updates at between 26.7 pJ and 111.25 pJ for an RTX 4090, and between 1.91 nJ and 47.75 nJ for a Raspberry Pi 5.

While the field-driven implementation for the NRAs used here has poor energy efficiency, the magnetisation dynamics exploited can be driven via spin-orbit torques. Based upon simulations in MuMax3 \cite{vansteenkiste_design_2014}, the current density required to drive domain walls similarly to magnetic fields is on the order of 4×10^{11} A/m with bias fields of 0.3 T, and can achieve clock speeds of around 100 MHz. For 25x25 ring arrays as used here, with 0.5 μm diameter rings shown to have equivalent behaviours [ref Allenspach 2024], this equates to 31.25 mW to drive the array. Encoding an input into a single rotation as used here, a single input costs around 313 pJ.

*Based upon these calculations, the energy efficiency of the NRAs sits between high-end devices optimised for machine learning (RTX 4090), and low-power devices capable of edge computing (Raspberry Pi 5). While this proves favourable for the edge applications these devices are most appropriate for, we would like to reiterate that the focus of this paper is on the methodology employed rather than the proposal of physical devices themselves. \ *

Recent studies into dynamic physical neural networks using spin-torque nano-oscillators (STNOs), widely considered the current state-of-the-art in spintronic computing, show suitability for dynamic PNNs in simulation, with energy estimates of around 100 fJ per neuron operation [ref Plouet 2025]. This makes STNOs a promising candidate for employing our methodology to realise these networks in hardware, with dramatic improvements in energy efficiency even compared to GPUs.

5- Please discuss the feasibility and potential challenges of scaling the employed PNN devices.

While we believe the critical advancements of our paper involve the proposed training methodology rather than the long-term applicability of the case-study devices we use to validate the methodology, to highlight these key concerns around scalability, some commentary around this area has been added to the discussion:

“While the dynamical systems studied here are promising candidates for neuromorphic computing, current approaches to signal input and state readout present practical challenges for large-scale implementation. For example, artificial spin-vortex ice requires high-precision, low-throughput measurement equipment, and applying magnetic fields is both slow and energy-intensive. Similarly, nanoring devices rely on electromagnets that consume much more energy than the underlying physical computation, and bridging electrical and magnetic domains adds further complexity.

Nonetheless, the methodology described here is broadly applicable to any dynamical system modeled by differential equations. This flexibility enables the optimization of networks based on alternative device platforms, where integration with existing CMOS technology may be more straightforward.”

6- Finally, please add a quantitative comparison between modeling with colored noise vs. a simpler i.i.d. white noise assumption. Providing a power spectral density (PSD) analysis or other metrics in the supplementary material could demonstrate the necessity of auxiliary variables in the Neural-SDE.

The authors agree that motivating the advantages of SDE should be clearer. For modelling purposes, we investigated the autocovariance structure of Neural-SDE reconstructions with the autocovariance structure of leaky integrators with noise terms with controllable timescales. This is shown in figure 7 in the supplementary material.

Additionally, we have added power spectral density figures for the raw experimental data, filtered experimental data, and the SDE's predictions. We have also provided plots of 'noise' by zeroing out the terms in the PSD plot associated with the AMR signal from the rings (128Hz), and its harmonics in all cases. It can be observed that the noise predominantly acts around the component of the signal (50-250 Hz) with some additional low frequency noise

that the filter does not attenuate without altering the signal (10 - 50 Hz). While the SDE replication is not perfect, it picks up on key features of the noise structure:

Figure 19: (a) Power spectral density plot of raw signal gathered from nanomagnetic ring arrays. Noise mainly centres around the frequency of the main component of the AMR signal (128 Hz) as well as some noise at lower frequencies (<50 Hz) (b) Resulting spectral density of experimental data after filtering with a third-order band-pass filter with corner frequencies of 50 and 200 Hz. Reduces the scale of low frequency noise, and noise beyond 200 Hz. (c) Power spectral density of the neural-SDE prediction of input waveforms used for panels (a) and (b). While the mapping is not perfect, the SDE model is able to reproduce qualitatively the key features of the plots in (a) and (b). Panels (d), (e), and (f) focus upon the lower magnitude frequency responses largely attributed to noise of the signals plotted in panels above.

To better motivate the neural SDE, we have repeated training of the MNIST task with Neural-ODEs with added gaussian noise equal in magnitude to experimental data (See figure 15a, supplementary material). This ‘noise’ magnitude value is generated by subtracting the Neural-ODE’s deterministic prediction from the data used to train the SDE (a repeated sampling of sequences of random input), and resembles the magnitude of noise but loses its autocorrelation structure. While this improved performance and negated training-transfer mismatch, the lack of noise-awareness in the backward pass meant the accuracies achieved were lower than the noise-tolerant configurations discovered via SDE:

Reviewer #3 (Remarks to the Author):

Manneschi et al. reported a study on a new method named noise-aware training to implement training strategy in physical neural networks (PNN). They developed a framework to training physical system that accounts for their dynamics and noise, extending beyond the previous Physics aware training (PAT) that can only train static physical devices. The authors

have demonstrated the methodology and results on modified MNIST dataset and Mackey-Glass prediction tasks. PNN is an emergent field that can accelerate the training efficiency of artificial neural networks. The authors present an interesting idea that allows one to extend the PNN training towards more dynamical physical systems. However, several critical issues and missing results need to be well-addressed prior to the publication:

1) This work share similarity with the previous PAT paper [Nature 601, 549 (2022)] that one needs to build a digital twin model to simulate a real physical system. Such method has a critical issue known as “simulation-reality gap” since it is always difficult to accurately include all degrees of freedom and complex interactions in a real physical system by a model. Such issue causes performance degradation during both the training and inference stages. When extending to time-dependent dynamical memory data processing, one can imagine such inaccuracy will be significantly amplified. Especially, the authors seem to model the spin textures by only adapting the input-output experimental results without considering the real spin dynamics. Therefore, it is puzzling whether this framework can adapt such inaccuracy. I understand that characterizing noise can improve such issue as presented in the manuscript, but the main issue here is whether a numerical model can effectively simulate a real complex physical system when time is also involved.

The authors appreciate the reviewer’s attention to the simulation–reality gap, which is indeed a critical consideration in modeling physical systems. We agree that simulating the underlying spin dynamics can provide valuable insights into the time evolution of spin textures. However, our approach is intentionally phenomenological: the goal is not to reproduce every microscopic detail of the spin system, but rather to accurately capture the device’s input–output response for the purpose of optimizing connectivity in physical neural networks. This strategy leverages the capacity of neural networks as universal function approximators.

In our framework, information is exchanged between neurons through experimentally accessible measurements, such as anisotropic magnetoresistance or spin-wave fingerprinting. While these observables are linked to the underlying spin structure, it is not necessary to resolve all spin degrees of freedom to predict the relevant device outputs. Instead, our models use current and recent inputs along with a short-term history of output states. As long as the history window is sufficiently long to treat the system as approximately Markovian, future states can be predicted from accessible measurements and inputs—bypassing the need for complete knowledge of all internal spin variables.

Our results, as shown in Figures 3 and 4 of the main text and Supplementary Figures 10 and 11, demonstrate that the models reliably reproduce previously unseen experimental data using only initial conditions and external input signals. Because the variables that mediate information exchange in the network are directly predicted by these models, we observe minimal simulation–reality gap for the key operational variables in the physical neural network.

To highlight these points in the main text, the following discussion has been added:

“In this work, we study two spintronic systems: nano-magnetic ring arrays (NRA) \cite{dawidek_dynamically-driven_2021,vidamour2022quantifying,vidamour2023reconfigurable,venkat2024exploring} and artificial spin vortex ice (ASVI) \cite{gartside2022reconfigurable,stenning2024neuromorphic}. As illustrated in Figure~\ref{fig:Fig2_scheme}(b) for the NRA, device responses exhibit stochastic variation across repeated presentations of the same input sequence, with the response distribution shaped by both current and past inputs. This variability originates from intrinsic physical dynamics and experimental noise. To address the resulting simulation-reality gap, we extend the Neural stochastic differential equations (Neural SDE) framework to capture signal-dependent noise with complex autocorrelation. Compared to approaches such as long short-term memory networks (LSTMs), neural differential equations provide several advantages: higher predictive accuracy, implicit access to partial derivatives via numerical integration, and natural integration of stochastic dynamics through SDEs. Supplementary Figure~\ref{fig:ODE_LSTM} compares the prediction accuracy and number of trainable parameters for Neural ODE and LSTM models.”

And

“Figure \ref{fig:Fig2_scheme}(c) shows the architecture of the proposed Neural-SDE model, comprising two neural networks: one for deterministic dynamics and another for stochastic dynamics. These networks are coupled via the numerical integration method, enabling the model to represent explicitly how system output depends on device state and external inputs, incorporating both deterministic and stochastic components. This structure supports noise-aware gradient computation for backpropagation through time (BPTT). The Neural-SDE architecture thus parameterizes the stochastic differential equations that define how device output evolves as a function of the current state and external input.\”

The deterministic network (upper) and the stochastic network (lower) each receive external input signals and a sequence of past device states. The history length must be sufficient to approximate the system as Markovian, ensuring that future states can be predicted from current inputs and recent device states. When this criterion is met, the method is applicable to any dynamical system. The stochastic network also receives auxiliary variables to support noise modeling. Outputs from both networks are integrated using a stochastic numerical scheme to produce the device’s activity (readout) at the next timestep. This value is recursively fed back as the most recent device state. Orange arrows in the figure indicate error gradients with respect to device activity and external input, as computed during BPTT.

2) The authors build two models, ODE and SDE to provide the deterministic and stochastic dynamics of the physical system, which is an interesting approach. However, this approach automatically decouples the correlation between the deterministic and stochastic (noise) signals since they are trained in separate networks. In real physical system, the level of noise signal is often correlated to the deterministic signal, either on its amplitude or time-dependent noise patterns. With such information missing, I doubt that one can guarantee the general performance of this training method.

The authors would like to thank the reviewer for highlighting that this important feature was not sufficiently clarified in the original manuscript. Our neural SDE model does capture the interaction between the deterministic signal and the noise due to the auto-regressive nature

and way that the NN functions in the SDE receive as input the previous state information (which will contain the deterministic signal). Further to this we have incorporated additional stochastic auxiliary variables which can induce longer timescale noise signals in the system dynamics. These connections are shown in figure 2c and supplementary figure 6. The changes made in response to the previous comment are also designed to address this initial oversight, and are repeated here for clarity:

“These networks are coupled via the numerical integration method, enabling the model to represent explicitly how system output depends on device state and external inputs, incorporating both deterministic and stochastic components. This structure supports noise-aware gradient computation for backpropagation through time (BPTT). the Neural-SDE architecture thus parameterizes the stochastic differential equations that define how device output evolves as a function of the current state and external input. .”

3) The author lacks comprehensive comparison on the results to the standard digital training methods, for example, reservoir computing, spiking neural networks which is aimed to treat similar tasks. An overall comparison of the relevant benchmarks is important (more than Fig. S12), i.e., number of layers, size of weights and accuracy etc.

We agree with the reviewer that comprehensive comparison is critical for evaluating performance. To provide better clarity, we have added the following text:

(For MNIST Task)

“The baseline performance, shown as red dashed lines, corresponds to physical neural networks with the same architecture but randomised hidden-layer connectivity, following the reservoir computing paradigm for both one-layer (left, orange) and two-layer (right, green) networks. The hidden weights are randomly drawn from distributions matched to those of the Neural-SDE-optimised network (see \ref{fig:Supp_perfBaseline_Rings}b), while only the output weights are optimised. This demonstrates that achieving high performance requires optimisation of the entire network connectivity, not just the output layer..”

(For Neuroprosthetics Task)

“Figure \ref{fig:Fig3}(d) shows the measured performance of a network whose connectivity was optimized in simulation using digital twins and then transferred to the physical system. As a baseline, we used a randomly connected network with a single hidden layer and trained only the output weights, following the standard reservoir computing (RC) approach, with the number of nodes matched to the PNN. This adjustment from previous baselines avoids the performance degradation seen with random connections in deeper networks while maintaining the same node count. The optimized network achieves an error rate approximately 30% lower than the baseline, demonstrating the advantages of optimized connectivity over conventional \emph{in-materio} RC and highlighting the improved capability made possible by connectivity optimization.”

(For Mackey-Glass signal prediction)

“Figure \ref{fig:Fig5}(b) shows the mean squared errors (MSE) between the ground truth dynamical equations and the predictions as a function of prediction steps into the future, for transferred networks with two (red) and three (blue) hidden layers. Cascade learning

achieves the lowest MSE, indicating excellent alignment between target and prediction, as illustrated in Figure \ref{fig:Fig5}(c). For reference, previous implementations of these experimental systems on this task reported a peak MSE of 3.86×10^{-2} at $t+5$ using the reservoir computing paradigm \cite{gartside_reconfigurable_2022}, and approximately 1×10^{-2} with multilayer PNNs trained without gradient-based optimization \cite{stenning2024neuromorphic}.

While spiking neural networks (SNNs) are indeed prominent in physical neuromorphic computing, their defining feature—a discontinuous spike event—complicates direct comparison. In this work, we focus on nonlinear leaky integrators, which underpin the activation dynamics in SNNs but exclude explicit spiking. This choice allows for a more immediate comparison with our physical neural networks.

For benchmarking, we provide results using nonlinear leaky integrators for each task addressed in this study. Details regarding the number of layers, parameter counts, and achieved accuracies are reported for each relevant task, as previously addressed in our earlier responses and summarized again here for clarity:

“Two-layer networks of nanorings optimised using interacting neural-SDEs demonstrate accurate transfer to physical devices and achieve performance exceeding that of static software-based MLPs with identical architectures (two hidden layers of 200 nodes). These networks also match the performance of dynamic MLPs incorporating leaky integrators (transferred accuracies for NRAs: 96.8\% at 25\% visibility and 96.0\% at 20\% visibility, versus 90.0\% and 88.0\% for static MLPs, and 96.9\% and 96.7\% for leaky-integrator MLPs; see Supplementary Figure 10). These results show that the magnetic nanoring PNN effectively exploits the intrinsic device dynamics as a memory resource, underscoring the capability of the proposed framework to leverage physical dynamics for in-memory computing. However, the limited memory of NRAs results in a relative decline in performance at lower visibility compared to software leaky integrator networks.”